# Fatal iatrogenic cerebral β-amyloid-related arteritis in a woman treated with lecanemab for Alzheimer's disease

Elena Solopova [1,9], Wilber Romero-Fernandez [1,9], Hannah Harmsen[2], Lissa Ventura-Antunes[1], Emmeline Wang[1], Alena Shostak[1], Jose Maldonado [3], Manus J. Donahue[1], Daniel Schultz [4], Thomas M. Coyne[5], Andreas Charidimou[6] & Matthew Schrag [1,7,8] ✉

We report the case of a 79-year-old woman with Alzheimer's disease who participated in a Phase III randomized controlled trial called CLARITY-AD testing the experimental drug lecanemab. She was randomized to the placebo group and subsequently enrolled in an open-label extension which guaranteed she received the active drug. After the third biweekly infusion, she suffered a seizure characterized by speech arrest and a generalized convulsion. Magnetic resonance imaging revealed she had multifocal swelling and a marked increase in the number of cerebral microhemorrhages. She was treated with an anti-epileptic regimen and high-dose intravenous corticosteroids but continued to worsen and died after 5 days. Post-mortem MRI confirmed extensive micro-hemorrhages in the temporal, parietal and occipital lobes. The autopsy confirmed the presence of two copies of APOE4, a gene associated with a higher risk of Alzheimer's disease, and neuropathological features of moderate severity Alzheimer's disease and severe cerebral amyloid angiopathy with perivascular lymphocytic infiltrates, reactive macrophages and fibrinoid degeneration of vessel walls. There were deposits of β-amyloid in meningeal vessels and penetrating arterioles with numerous microaneurysms. We conclude that the patient likely died as a result of severe cerebral amyloid-related inflammation.

Over the last 15 years, numerous clinical trial programs have tested infusions of monoclonal antibodies against various forms of β-amyloid in an effort to facilitate clearance of plaques, oligomers, and/or soluble β-amyloid peptides from the brains of patients with Alzheimer's disease. This approach emerged on the heels of a promising, although ultimately unsuccessful, effort to vaccinate patients with Alzheimer's disease against β-amyloid. The trial of the AN1792 vaccine to induce active immunotherapy ended in early 2002 after a subset of patients developed encephalitis[1]. Neuropathologic characterization of several of those patients showed they had accelerated cerebral amyloid angiopathy leading to a breakdown of the blood–brain barrier, inflammatory infiltration, and bleeding[2]. The underlying pathophysiology of this side-

[1]Department of Neurology, Vanderbilt University Medical Center, Nashville, TN, USA. [2]Department of Pathology, Vanderbilt University Medical Center, Nashville, TN, USA. [3]Vanderbilt Neurovisualization Lab, Vanderbilt University, Nashville, TN, USA. [4]Final Diagnosis: Private Autopsy Florida - Forensic Pathology Lab, Tampa, FL, USA. [5]Department of Pathology, Immunology and Laboratory Medicine, University of Florida, Gainesville, FL, USA. [6]Department of Neurology, Boston University, Boston, MA, USA. [7]The Vanderbilt Brain Institute, Vanderbilt University, Nashville, TN, USA. [8]Vanderbilt Memory and Alzheimer's Center, Vanderbilt University Medical Center, Nashville, TN, USA. [9]These authors contributed equally: Elena Solopova, Wilber Romero-Fernandez. ✉e-mail: matthew.schrag@vanderbilt.edu

effect suggests that intramural periarterial drainage along the walls of capillaries and arteries was overwhelmed. These findings in β-amyloid vaccination were similar to pathological features seen in the spontaneous disorder known as cerebral amyloid angiopathy-related inflammation (CAARI)[3]. With passive antibody administration, a similar clinical and neuroimaging phenomenon has been observed and termed ARIA or β-amyloid-related imaging abnormality. This side-effect is common, occurring in up to 40% of patients in some trials, although most cases have been asymptomatic[4]. The neuropathological features of ARIA are not well understood owing to few cases in the published literature, and consequently, there is some debate about whether the mechanism of ARIA is equivalent to the earlier encephalitis cases and to CAARI[5,6]. Here, we report the clinical, neuroimaging, and neuropathological features of a fatal case of ARIA associated with infusions of the anti-β-amyloid monoclonal antibody lecanemab (originally known as BAN-2401). We conducted post-mortem neuroimaging and histological studies to evaluate the tissue changes associated with this syndrome. The patient's surrogate decision-makers provided informed consent for autopsy, tissue donation for scientific study, and publication.

## Results

### Brief case history

The patient was a 79-year-old woman in good general health, with no clinical history of hypertension or neurological disorder except for an approximately four-year history of mild but progressively worsening memory symptoms which did not impair her functional independence. She was enrolled in a randomized controlled trial of lecanemab where she was assigned to the placebo group. The patient reported no adverse effects or perceptible benefits. At the completion of the trial, she was screened with magnetic resonance imaging (MRI), Tau and β-amyloid positron emission tomography (PET) scans, which supported the diagnosis of Alzheimer's disease, and she was enrolled in the open-label extension, which guaranteed treatment with the active drug. She was treated with three infusions of lecanemab, each dose (10 mg/kg) approximately 2 weeks apart. According to her study partner, approximately an hour after each dose she developed a headache causing her to spend 1 to 2 days in bed recovering each time. After the third dose, she began to experience progressively worsening memory impairment which she described as brain fog. On the day of hospital admission, she had a seizure that began with the left head and gaze version and left-sided tonic contraction which evolved into 30 s of generalized convulsion. She regained alertness after this event but was never communicative nor purposefully interactive again. Electroencephalography obtained on the hospital day 1 did not suggest ongoing seizures to explain her condition, although she had frontal intermittent rhythmic delta activity. Neuroimaging on hospital day 2 revealed a vasogenic pattern of cerebral edema in the bilateral temporal, parietal, and occipital lobes with numerous microhemorrhages. She was treated with 1 g daily intravenous solumedrol for 3 days for suspected amyloid-related imaging abnormality (ARIA) without improvement. She was also treated with a heparin infusion for atrial fibrillation discovered on her arrival at the hospital. Her neurological condition did not improve, and, on hospital day 4, she suffered an aspiration event leading to sepsis with multiorgan failure. Consistent with her wishes aggressive resuscitation measures were not instituted. She died 5 days after hospital admission. An extended case history is included in the supplementary material.

### Neuroimaging

A pre-treatment MRI was obtained as part of the enrollment procedure before the open-label extension component of the clinical trial and a post-treatment MRI was obtained as part of her clinical care after her seizure and neurological deterioration. The pre-treatment study showed moderate white matter disease without mass effect on FLAIR sequences (Fig. 1A and Supplementary Fig. 1). This is grade 2 white matter disease on the age-related white matter changes (ARWMC) scoring scale[7]. The presence of cortical microhemorrhages and this pattern of white matter disease meets the Boston criteria for a diagnosis of probable CAA, both by the original and recently updated sets of criteria[8,9]. The post-treatment study obtained on hospital day 2 shows expansion of the white matter signal abnormality with edema in both temporal lobes, parietal lobes, and occipital lobes (Fig. 1 and Supplementary Fig. 2). The pre-treatment study showed four small cerebral microhemorrhages on gradient echo-T2* (arrows in Fig. 1B and supplementary Fig. 3). The post-treatment study shows more than 30 microhemorrhages, all in a cortical or juxtacortical distribution and most within areas of edema (Fig. 1D and Supplementary Fig. 4). Selected images from the diffusion-weighted and apparent diffusion coefficient (ADC) scans associated with her hospital admission are shown in Supplementary Fig. 5; there was no major stroke. Post-contrast images showed no contrast enhancement. Before initiating treatment with lecanemab in the open-label extension study, the patient underwent a β-amyloid PET scan with florbetaben and a tau PET scan with flortaucipir. Representative images from these studies are shown in Supplementary Fig. 6. Tracer retention suggested both amyloid and neocortical tau positivity at entry into the open-label extension phase of the trial.

The post-mortem brain was scanned separately at the clinical field strength of 3T and at the high field strength of 7T to further evaluate the extent of hemorrhagic changes in the brain (Fig. 1E, F, G, H, respectively). As expected, the hemorrhagic changes visualized on the post-mortem imaging are much more extensive than on the in vivo scan. To confirm the extent of microhemorrhagic changes, we subjected a 0.5 cm thick section of a gyrus from the temporal cortex to optical clearing and when it was translucent at roughly the midpoint during the clearance procedure, we photographed the block with backlighting to show the full extent of bleeding (Fig. 1I, J). The density of microhemorrhages visualized pathologically and on post-mortem MRI is higher than was visualized on the clinical in vivo MRIs.

### Neuropathology

On gross inspection of the brain, there was evidence of edema in the temporal, parietal, and occipital lobes based upon flattening of the cortical gyri against the leptomeninges. Petechial hemorrhages were visible on the natural and cut surfaces of the brain (Fig. 2A–C). Sixteen blocks of tissue were prepared for histological evaluation (shown in Supplementary Fig. 7). Immunohistochemistry for β-amyloid and phosphorylated tau demonstrated the presence of neuritic plaques and neurofibrillary tangles consistent with intermediate Alzheimer's disease neuropathologic changes (Thal phase 5 of 5 for β-amyloid deposition, Braak and Braak stage IV of VI for neurofibrillary tangles, and CERAD frequent neuritic plaques: Intermediate Alzheimer's disease Neuropathology Change A3, B2, C3). To further characterize the plaques, we stained sections with methoxy-X04 (a sensitive blue fluorescent probe for amyloid and aggregated tau), vacuolar ATPase subunit V0A1 or phospholipase D3 (two of the most sensitive markers of dystrophic neurites we have found to date)[10] and IBA1 (a marker of microglia). While most of the plaques looked entirely typical, 21% appeared to have been "cleared" – which is to say, a distinctive rosette of dystrophic neurites was present without the typical amyloid staining at its center. Further, 24% of the plaques had minimal staining of amyloid deposits (Supplementary Fig. 8). These features were associated in some cases with intense immunoreactivity for IBA1/microglia (Fig. 2D). Numerous areas of microhemorrhage were visualized histologically along with arterioles with varying degrees of fibrinoid necrosis and perivascular inflammation. The bulk of the perivascular inflammation was composed of macrophages with occasional multinucleated giant cells (stained by CD68); some T-cells were also present in the perivascular inflammatory milieu (Figs. 2E–I and 3). CD68 staining of macrophages and activated microglia were present in

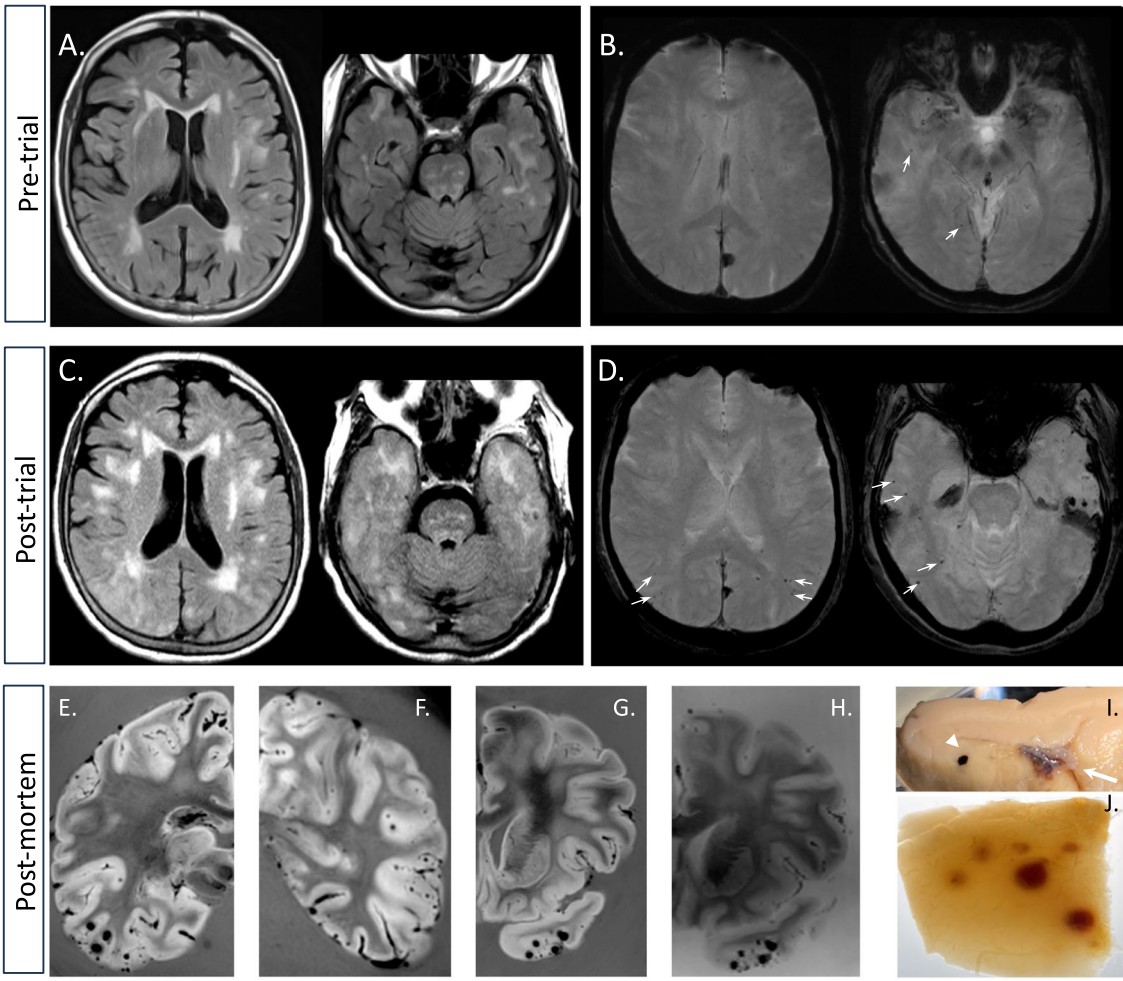

**Fig. 1 | Neuroimaging evidence of cerebral edema and microhemorrhage. A** Pre-trial FLAIR shows pre-existing moderate white matter disease (a montage of additional MR images available in the Supplementary Fig. 1). **B** Susceptibility weighted minimum intensity projection imaging revealed four cortical microhemorrhages (two are visible in the selected images at the arrows; the others are marked in Supplementary Fig. 3). **C** Hospital-acquired FLAIR MR image shows increased hyperintensity, suggesting exacerbated white matter disease, and sulcal efface-ment/mass effect in the temporal, parietal, and occipital lobes, suggesting the emergence of edema after the trial (a montage of additional MR images available in the Supplementary Fig. 2). **D** Hospital-acquired gradient echo-T2* minimum intensity projection MR image revealed that the number of microhemorrhages increased to over 30 (eight are shown at the arrows; the rest can be seen in Supplementary Fig. 4). **E**–**G** Post-mortem susceptibility weighted imaging at 3 Tesla demonstrated extensive microhemorrhagic changes, most prominently in the temporal, parietal, and occipital lobes. **H** 7-Tesla MR imaging, paired with G shows that additional microhemorrhages are detected at higher field strength. **I** A few microhemorrhages (arrowhead) and superficial siderosis (arrow) were visible on gross sections. **J** More extensive microhemorrhagic changes were readily observed when the tissue was rendered translucent with optical clearance approaches and backlighting.

the leptomeninges and involved some areas of parenchyma, especially near involved vessels. Areas with less prominent edema on the neu-roimaging, including most of the frontal lobes, had less prominent perivascular immunoreactivity for CD68. The patient also had severe cerebral amyloid angiopathy. The ruptured arterioles associated with microhemorrhages contained significant β-amyloid deposits (Fig. 2J, K). Three-dimensional views of degenerating and ruptured arterioles were obtained from light-sheet microscopy of cleared tissue and are shown in Supplementary Movies 1 and 2. In many cases, blood extended from the site of arteriolar rupture within contiguous peri-vascular spaces, as was previously described (Figs. 2K and 3)[11]. Vascular amyloid deposition, periarterial inflammation, and microaneurysms were also frequently observed in meningeal specimens (Fig. 2L–N). APOE genotyping was not available from her clinical records, so this was obtained as part of her autopsy; she carried two copies of the E4 allele.

## Discussion

This case shows the neuroimaging and neuropathological features of an acute arteritis that occurred during the lecanemab treatment,

an experimental anti-β-amyloid therapy for Alzheimer's disease. The clinical, pathological, and neuroimaging findings in this case are consistent with a severe form of a known side-effect of this class of drugs which has been termed amyloid-related imaging abnormality, or ARIA[12]. Neuropathologically, we found that her condition was associated with marked perivascular inflammation and arteriolar degeneration resembling fibrinoid necrosis, leading to micro-hemorrhagic changes in both the parenchyma and leptomeninges. The inflammatory features include extensive macrophages and/or activated microglia in the leptomeninges, perivascular space, and adjacent parenchyma, with occasional multinucleated giant cells in vessel walls. These features were associated with severe cerebral amyloid angiopathy. The patient's neuroimaging along with this constellation of neuropathology findings resembles the sporadic condition cerebral amyloid angiopathy-related inflammation (CAARI). The neuroimaging features of CAARI that are present in this case include multifocal cerebral edema with cortical microhemor-rhages; contrast enhancement is an inconsistent feature of CAARI and is absent here. The inflammatory side effects in this case were

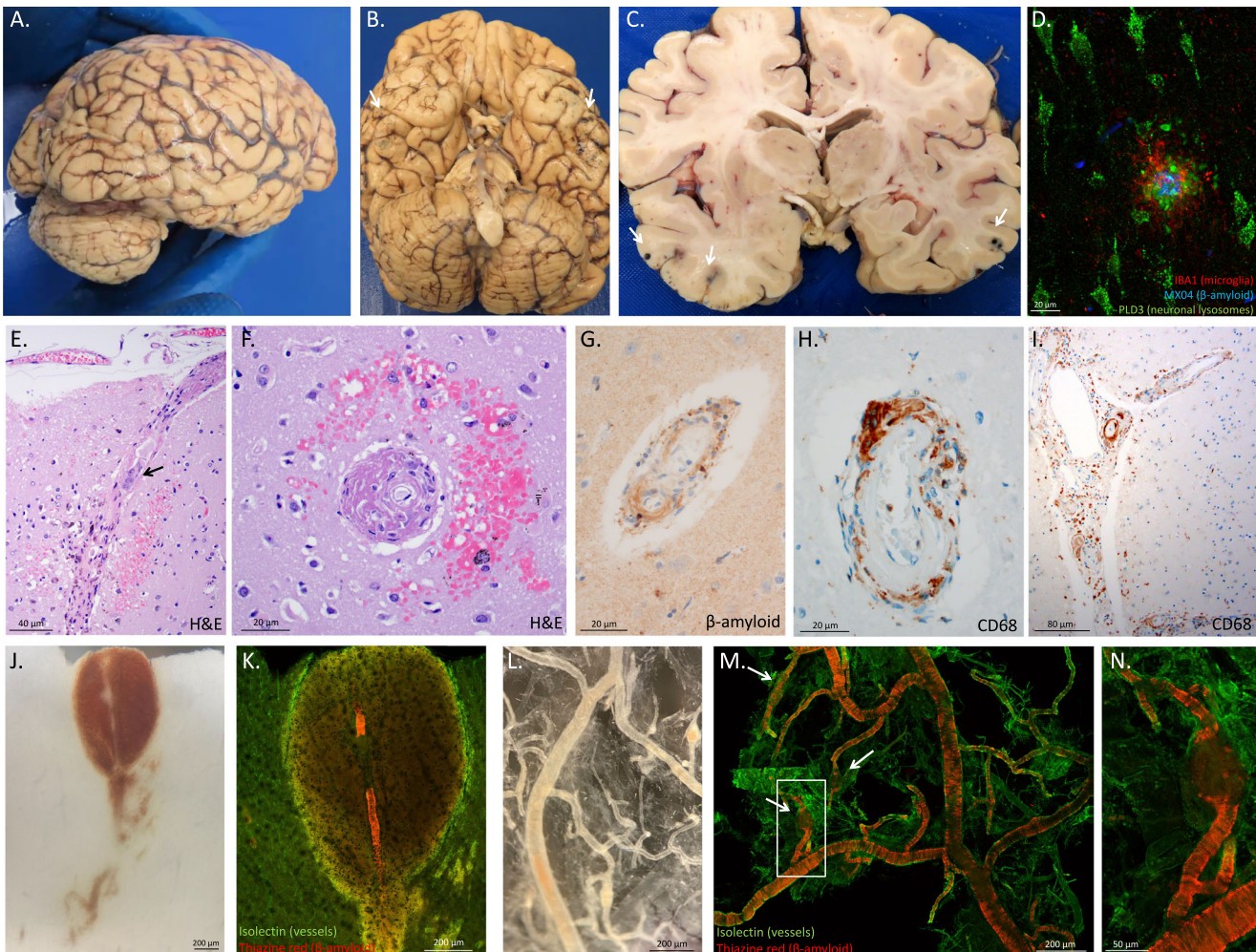

**Fig. 2 | Inflammatory and microhemorrhagic pathology. A** Lateral and **B** inferior views of the brain. Black stippling (arrows) in the temporal lobes consistent with petechial hemorrhage is best seen on the inferior view. **C** A coronal section through the brain shows additional microhemorrhages (arrows). **D** Methoxy-X04 staining of β-amyloid (blue), phospholipase D3 (green, staining neuronal lysosomes and dystrophic neurites) and Ionized calcium Binding Adaptor molecule 1 or IBA1 (red, staining microglia) shows a neuritic plaque surrounded by dystrophic neurites and activated microglia, 20 μm scale bar. This case met the criteria for moderate Alzheimer's disease neuropathologic changes. Additional images of β-amyloid deposits in this case are shown in Supplementary Fig. 8. **E** Longitudinal and **F** cross-sectional images of penetrating arterioles in the affected region show intensive perivascular inflammatory infiltration and bleeding, stained with hematoxylin and eosin (H&E), 40 μm scale bar in **E** and 20 μm scale bar in **F**. A multinucleated giant cell is present at the arrow in **E**. **G** Immunostaining for β-amyloid demonstrates inflamed vessels have β-amyloid deposition, 20 μm scale bar. **H** Cross-sectional and I. longitudinal sections were immunostained for CD68 (Cluster of Differentiation 68), showing numerous macrophages on the inflamed vessels (20 μm scale bar in **H** and 80 μm scale bar in **I**); see also Fig. 3. **J** Serial sectioning through a micro-hemorrhage enabled visualization of the associated ruptured vessel, 200 μm scale bar. **K** The ruptured vessel associated with the microhemorrhage contains heavy β-amyloid deposits stained red, vascular marker isolectin in green, blood in non-specific yellowish autofluorescence around the vessel, 200 μm scale bar. **L** Meningeal whole-mount specimen overlying temporal lobe shows several microaneurysms, 200 μm scale bar. **M** Meningeal specimen from **L** stained with thiazine red for β-amyloid (red) and isolectin to label vessels (green) shows severe β-amyloid deposition, including in several microaneurysms (arrrows), 200 μm scale bar. The aneurysm in the white box is shown enlarged in **N** with a 50 μm scale bar.

refractory to treatment. In sporadic CAARI, most cases respond well to corticosteroid treatment, but a minority are refractory and may have poor outcomes. Autopsy results from another case also enrolled in the open-label extension of the same trial were remarkably similar to our findings, with intense perivascular inflammation and microvascular degeneration. In that case, the patient died after hemorrhagic complications from administration of a fibrinolytic agent for an acute stroke, highlighting the potential hemorrhage risk[5,6]. These two cases are the only detailed neuropathological descriptions of active ARIA in the existing literature. Two additional cases from an earlier trial of bapineuzumab were evaluated long after episodes of ARIA had resolved; both of these cases had severe CAA, but no evidence of active inflammation[13,14].

This patient's first symptom was a headache occurring shortly after the initiation of each infusion. Headaches were reported in 11.1% of patients treated with lecanemab, compared to 8.1% in the placebo arm; the headache rates were similar in the aducanumab and dona-nemab trials[4,15,16]. It is unclear if the presence of a headache correlated with the risk of cerebral edema. We are not aware of prior reports of patients experiencing headaches timed right after infusions of anti-β-amyloid immunotherapy agents, thus it is unclear whether the timing of the headaches with infusions represents a higher risk for inflammation. This relationship could be clarified through a retrospective analysis of recent clinical trials.

This case highlights an important side-effect of the anti-amyloid immunotherapy drug class. Two out of 1895 patients in the phase III CLARITY-AD trial of lecanemab had neuropathologically documented cerebral inflammation in the spectrum of CAARI and approximately 1% of patients had severe cerebral edema by radiographic criteria. Such frequency is higher than the reported rate of

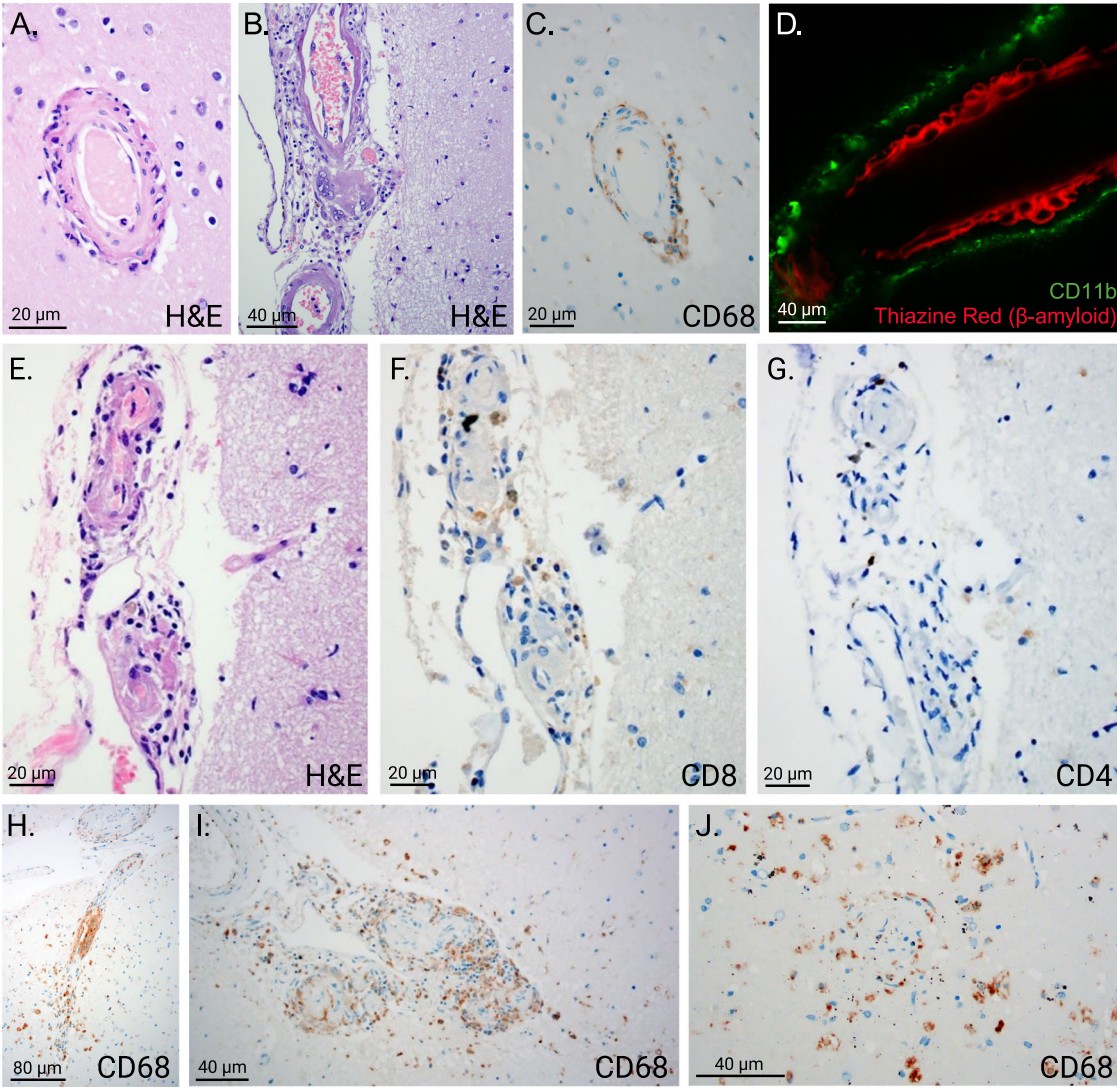

**Fig. 3 | Characterization of cell types in perivascular inflammation.**
**A** Hematoxylin and eosin staining (H&E) demonstrates inflammatory perivascular infiltrates, 20 μm scale bar. **B** Multinucleated giant cells are visible in this image, 40 μm scale bar. **C** CD68 (CD stands for Cluster of Differentiation) immunostaining shows that many of the inflammatory cells are macrophages and/or activated microglia, 20 μm scale bar. **D** CD11b, a marker of microglia and macrophages, shows numerous macrophages in the perivascular space around a vessel with heavy β-amyloid deposition (shown by red staining with Thiazine red), 40 μm scale bar. Images **E**–**G** show that the perivascular inflammatory infiltrate also contains T-cells, although they are less abundant than macrophages, 20 μm scale bars. Image **E** shows hematoxylin and eosin staining of an inflamed vessel, **F** CD8 staining and G. CD4 staining of the same vessel. Images **H**–**J** show patterns CD68 immunoreactivity which involve the leptomeninges (**H** & **I**), perivascular spaces (**H** & **I**), and parenchyma (all three images), 80 μm scale bar in **H** and 40 μm scale bars in **I** and **J**.

spontaneous CAARI which is estimated to occur in 0.13/100,000 people, making it unlikely that these cases were incidental occurrences of sporadic CAARI[15,17]. Around 2010, the FDA recommended restricting patients with two or more microhemorrhages from participating in clinical trials of anti-amyloid immunotherapies due to the risk of developing a syndrome of vasogenic edema and/or microhemorrhages[12]. However, an academic/industry collaborative working group was assembled by the Alzheimer's Association to address this concern and advocate for less restriction in patient enrollment criteria[18,19]. This group introduced the term ARIA in 2011 to describe the syndrome, adding the modifier -E for edema or effusion and the modifier -H for the presence of hemorrhage[20]. Although around 75% of patients who develop ARIA remain asymptomatic and most of the remainder have a relatively benign course, severe cases like this one do occur. We suggest an alternative term be developed to describe ARIA when it is symptomatic or otherwise severe to more-accurately convey that it is not uniformly limited to an imaging feature[21,22]. The risk of developing inflammatory changes

after anti-amyloid immunotherapy correlates with the presence of CAA and associated risk factors, including APOE4 genotype[20,23]. It is also notable that patients who were homozygous for APOE4 did not clearly benefit from treatment in subgroup analyses in both the recent lecanemab and donanemab phase 3 trials[15,16]. Given the increased risk and lack of a clear benefit, it may be reasonable to avoid treating patients carrying two copies of the E4 genotype with this medication.

In this case, the patient met the Boston criteria for a diagnosis of probable CAA based on her pre-treatment MRI (based on either the original or the updated criteria). As expected, the in vivo MRI significantly underestimated the extent of microhemorrhages, confirmed by post-mortem imaging with optimized acquisition settings and on neuropathological inspection. Microhemorrhages on the clinical scan are certainly useful diagnostically but should be viewed as a marker of microvascular disease rather than an indicator of the extent of microvascular disease. The trial exclusion criteria permit enrollment of patients with up to 4 microhemorrhages. These criteria do not

consider the distribution of microhemorrhages and do not standardize the screening MRI protocols, and thus will likely lead to heterogeneity in CAA detection. The established detailed MRI-based diagnostic criteria for CAA are likely to perform better than the coarse criteria employed as exclusion criteria in this and many other clinical trials[8,9]. The difficulty with reliable detection of CAA is likely to be magnified as this treatment moves to the community setting where less sensitive sequences like gradient echo-T2* and lower field strength imaging are prevalent; therefore, more patients with co-morbid CAA are likely to meet the inclusion criteria. Another important safety consideration is that while patient screening and treatment could be restricted to facilities with adequate and standardized diagnostic capabilities, the treatment of the side effects is likely to occur in the community setting in many cases. At present, ARIA has rarely been encountered outside of a clinical trial population, so community health-care providers would likely benefit from education on the features and risks associated with this side-effect. Additionally, it is important moving forward that minimum imaging standards ensure robust screening for cerebral amyloid angiopathy. We would propose that all prospective patients undergo a screening MRI at a minimum of 3T field strength, with susceptibility-weighted imaging at a slice thickness of not more than 5 mm and no interslice gap. Patients meeting diagnostic criteria for probable CAA by the Boston criteria (2.0) are most likely not good candidates for treatment.

This case demonstrates an important gap in our understanding of the molecular mechanisms of ARIA. It would be tempting to interpret this inflammatory condition as an antibody-mediated autoimmune attack against β-amyloid in the cerebral microvasculature. However, ARIA rates in the clinical trials with aducanumab were also quite high, despite reports that aducanumab preferentially binds parenchymal versus vascular β-amyloid[24]. An alternative interpretation is that the induction of β-amyloid clearance mediated by these antibodies traffics much of the β-amyloid through the perivascular spaces, leading to periarteriolar inflammation that can worsen CAA. If this perivascular clearance overwhelms the capacity of the microvasculature, blood–brain barrier dysfunction may ensue initiating an inflammatory cascade. The latter interpretation is supported by animal studies[25,26]. Defining and mitigating the molecular mechanism of this side-effect may be key to improving the safety of anti-amyloid therapies.

Previous reports have described the appearance of plaques during and after clearance from immunotherapies. In this case, even after only three doses of lecanemab we noted several of the β-amyloid plaques appeared to have been cleared or in the process of clearance. However, this case was too early in the process of treatment to assess how other neuropathological features of Alzheimer's disease change with β-amyloid lowering therapies. Dystrophic neurites were still present around cleared plaques at this early timepoint. It will be important to determine whether dystrophic neurites resolve or are persistent, whether the various tau pathologies appear to be halted, whether granulovacuolar degenerating bodies remain, and whether brain atrophy is reduced after long-term treatment. Neuroimaging results would suggest brain atrophy is not halted (at least acutely)[27]. The pattern of change in these neuropathological features will have considerable bearing on whether minor alterations in the cognitive trajectory of treated patients are likely to represent disease modification or purely symptomatic effects[28].

In conclusion, this case should prompt a careful evaluation of the safety of lecanemab treatment and a refinement of the approach to pre-screening potential recipients of lecanemab for cerebral amyloid angiopathy and its risk factors. We advocate that patients should be screened for cerebral amyloid angiopathy using the Boston criteria and undergo APOE genotyping prior to treatment. The case also highlights an urgent need for neuropathological studies to define the cellular and molecular features of ARIA.

## Methods

### Inclusion and ethics statement
We confirm that our research complies with all pertinent ethical regulations, including CARE guidelines and the Declaration of Helsinki principles. The patient's surrogates provided informed consent to participate in this case report. Vanderbilt University Medical Center's Institutional Review Board provided ethical oversight as part of our ongoing Observational Study of CAA and Related Disorders (OSCAAR), protocol number 180287.

### Post-mortem MR imaging
Coronal tissue slabs were sliced coronally and embedded in 1% agarose under degassing conditions, with care to avoid air bubbles. Post-mortem susceptibility weighted imaging (SWI) and T2-weighted FLuid Attenuated Inversion Recovery (FLAIR) MRI were obtained on both 3.0 T (Philips Ingenia) and 7.0 T (Philips Achieva) human clinical scanners using 32-channel phased array reception. 3.0 T SWI (technique = 3D gradient echo, number of echoes = 4, first echo = 7.2 ms, echo spacing = 6.2 ms, repetition time = 31 ms, spatial resolution = $0.6 \times 0.6 \times 2$ mm), 3.0 T FLAIR (technique = 3D turbo-inversion-recovery, echo time = 271 ms, repetition time = 4800 ms, inversion time = 1650 ms, spatial resolution = $1.0 \times 1.0 \times 1.0$ mm), 7.0 T SWI (technique = 3D gradient echo, number of echoes = 9, first echo = 8.0 ms, echo spacing = 4 ms, repetition time = 64.3 ms, spatial resolution = $0.65 \times 0.65 \times 0.9$ mm, and 7.0 T FLAIR (technique = 3D turbo-inversion-recovery, echo time = 280 ms, repetition time = 3952 ms, inversion time = 1375 ms, spatial resolution = $0.80 \times 0.80 \times 0.80$ mm) were performed on the same day with sequence parameters chosen to parallel in vivo protocols.

### Immunohistochemistry
Immunostaining of brain sections for the autopsy was conducted according to Clinical Laboratory Improvement Amendments (CLIA) standards using approved antibodies. For exploratory analysis, floating sections were prepared on a Leica 1200 S vibratome with a thickness of 50 μm or 100 μm depending on the application. Meningeal tissue was gently removed from the surface of the brain, stained, and mounted as a whole-mount preparation. Aggregated tau and β-amyloid were detected using thiazine red (1 μM, Chemsaves) or methoxy-X04 (1 μM, Tocris, Minneapolis, MN), lectin (dilution 1:250, *Lycopersicon esculentum* (Vector Laboratories, Tomato) labeled, Fluorescein (FL-1171-1)) was used as a vascular marker. The primary antibodies we used were anti-PLD3 rabbit polyclonal antibody (3 μg/mL, HPA012800; Sigma-Aldrich, St. Louis, MO), anti-ATP6V$_0$A$_1$rabbit polyclonal antibody (10 μg/mL, NBP1-89342, Novus Biologicals), anti-CD11b chicken polyclonal antibody (10 μg/mL, MAC, AvesLabs) and anti-IBA1 mouse monoclonal antibody (5 μg/mL, Cat# 66827-1-Ig, CloneNo. 1C6A10, Proteintech, Rosemont, IL). The antibodies used in autopsy are the following: β-amyloid- ready to use prediluted 7 mL from BioSB, REF BSB 3442, clone RBT-A4, antibody incubation time 15 min, anti-Tau-1:15,000 from Invitrogen (Thermo Fisher Scientific), REF MN1020, clone AT8, antibody incubation 30 min; anti-CD68- ready to use from Roche, REF 790–2931, clone KP-1, antibody incubation 16 min, anti-CD4- ready to use from Leica, REF PA0427, clone 4B12, antibody incubation for 15 min, anti-CD8- ready to use from Leica, REF PA0183, clone 4B11, antibody incubation for 15 min with ER1 for 15 min. For the β-amyloid, Tau, CD4, CD8 antibodies, the following detection kit was used: Leica Bond III IHC stains- BOND Polymer Refine Detection- REF DS9800. For CD68 antibody, Ventana Roche Ultra Benchmark IHC stain- ultraView Universal DAB Detection Kit- REF 760-500 was used. Confocal images were acquired through the Vanderbilt Cell Imaging Shared Resource (CISR) using a Zeiss LSM 710 confocal laser-scanning microscope (Carl Zeiss AG, Germany) with a 20× air/dry or 63× oil-immersion objective with a minimum resolution of 2000 × 2000 pixels.

## CLARITY and light-sheet microscopy

For three-dimensional, light-sheet microscopy tissue blocks were incubated for 3 days in 4% paraformaldehyde (PFA) and embedded in CLARITY acrylamide hydrogel (4% Acrylamide, 0.05% Bis-Acrylamide, 0.25% temperature-triggering initiator VA-044 in 0.1 M phosphate-buffered saline in water- PBS) for 4-5 weeks. The tissue was then placed in a 37 °C water bath to initiate hydrogel polymerization. The blocks were passively clarified in 200 mM boric acid, 4% w/v SDS, pH 8.5 at 37 °C until translucent, then pigmentation was photo-cleared with exposure to an LED microarray. The blocks were stained with Lectin (*Lycopersicon esculentum* (Tomato) labeled, dilution 1:200, FL-1171-1, Vector Laboratories) and thiazine red (1 μM, Chemsaves #2150336) in PBS- Triton 0.1%. After overnight incubation in 68% thiodiethanol (TDE) and achieving a refractive index of 1.33, the blocks were imaged using a Light-Sheet microscope. Image processing was done with Imaris Microscopy Image Analysis Software (Oxford Instruments).

## Statistics and reproducibility

The nature of this case report suggests a detailed analysis of one patient's data. Thus, no statistical method was used to predetermine sample size. Due to the descriptive nature of the case report, no data was excluded, the experiments were not randomized, and the experimenters were not blinded to allocation during the experiment and outcome assessment. For the light microscopy from the patient's autopsy, histological and immunohistological findings are representative of an examination of the sixteen tissue blocks shown in Supplementary Fig. 7. For fluorescence microscopy, each marker was assessed on a minimum of five tissue sections. Three meningeal specimens were prepared.

## Reporting summary

Further information on research design is available in the Nature Portfolio Reporting Summary linked to this article.

## Data availability

All shareable data generated or analyzed during this case report has been made available in this manuscript and its supplementary materials. To deidentify MR images, sensitive patient data has been removed and is not publicly available to protect their privacy. The source files for the figures are available from the corresponding author upon reasonable request.

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

## Acknowledgements

We are grateful to the family of this patient who contributed her tissue and medical records for scientific study. This case study was made possible by philanthropic support from the family and friends of

Louis Stephen Zuzga Moran and the family and friends of Douglas B. Janney, Jr.

## Author contributions

E.S. and M.J.D. were responsible for the post-mortem MRI and histological analyses, additionally E.S. wrote a draft of the manuscript; W.R.F., E.W., E.S. were responsible for immunohistochemistry; H.H., D.S., and T.M.C. conducted pathological autopsy evaluation, L.V.A. and J.M. performed light-sheet microscopy and image rendering, A.S. contributed to the histology, tissue handling, and editing the final manuscript, A.C. and M.S. were responsible for analyzing clinical records. M.S. co-wrote the manuscript and supervised all aspects of the study.

## Competing interests

M.J.D. receives research-related support from Philips North America; is a paid consultant for Pfizer Inc, Alterity, Global Blood Therapeutics, Graphite Bio, and LymphaTouch; is a paid advisory board member for Novartis and Bluebird Bio; receives research funding from Pfizer Inc; and is the President/CEO of Biosight, LLC, which operates as a clinical research organization. A.C. reports receiving funding from the Bodossaki Foundation and the Frechette Family Foundation and consulting fees from Imperative Care. M.S. reports receiving funding from the American Federation of Aging Research, the National Institutes of Health, and consulting fees from Labaton-Sucharow LLP and Raymond James. The other authors declare no competing interests.
