## [Peer Review File · Nature Communications]

Fatal iatrogenic cerebral β -amyloid-related arteritis in a woman treated with lecanemab for Alzheimer's diseaseEditorial Note: Parts of this Peer Review File have been redacted as indicated to remove third-party material where no permission to publish could be obtained.

REVIEWER COMMENTS

Reviewer #1 (Remarks to the Author):

Dear Editor,

The article “Fatal Iatrogenic Cerebral Amyloid-Related Encephalitis in a patient treated with lecanemab for Alzheimer’s disease” is a case report of a fatal complication of the recently FDA-approved anti-amyloid immunotherapy, lecanemab, as part of the Open-Label Extension of the pivotal phase 3 Eisai-sponsored randomized-controlled clinical trial (RCT). This case report from a very productive and well-renowned group in the field of cerebral amyloid angiopathy (CAA), includes a thorough neuropath examination, and clinical and neuroimaging data. The results are simply and clearly presented. The evidence is convincing, and the findings and the discussion are of utmost importance for the use of this drug in light of the current FDA discussion regarding full approval of lecanemab and its use in clinical practice, especially in APOE4 homozygous carriers and in individuals with CAA. Regarding the scientific and medical aspects, I would simply argue against the emphasis on the term encephalitis by the authors and propose the use of vasculitis or angiitis instead. I am not aware of the space and reference limitations, but I would also recommend a few additional details and discussion necessary to put these findings into perspective. I am particularly surprised that the authors do not mention the recent similar report of the Chicago group concerning a case of lecanemab-induced vasculitis (DOI: 10.1056/NEJMc2215148 ; DOI: 10.3233/JAD-221305). I am not an expert in legal and political aspects, but I also have some concerns. I am not aware of exactly which physicians were involved in the management of the patient. However, from the Press, it seemed that the acute clinical event detailed in this case report was managed in a Floridian hospital. I am surprised that only Floridian pathologists and not ICU physicians or neurologists involved in the management of the patient are co-authors of this paper. I understand that some political dimensions are probably at-stake. Nonetheless, the Editor and the Authors should clarify this, for transparency. Finally, some data seem to have been collected during the Eisai-sponsored RCT (pre-infusion MRI, Amyloid, and Tau PET). We are also not sure who the deceased patient's legal representative was, and who consented to this publication. The agreement of the patient legal representative and the intellectual property of these data should be carefully reviewed by the Journal’s legal department to ensure the article will not be retracted after publication despite its medical and scientific importance and rigor. Note that I have no neuropathological training and could not review these aspects of the current manuscript in detail.

Major concerns

1. I agree with the authors when they propose a new acronym and medical term to describe these complications under anti-amyloid immunotherapies: the previously proposed ARIA (Amyloid Related Imaging Abnormalities) tends to undermine the severity of these events. Nonetheless, the use of encephalitis is debatable and can be confusing since the neuropathological description of the fatal event is mainly that of a drug-induced central nervous system vasculitis/angiitis. Indeed, the parenchymal microglial activation observed here around plaques was previously reported in autopsies of patients treated with anti-amyloid drugs, but who did not experience ARIA (DOI: 10.1007/s00401-022-02433-4 ;

DOI: 10.1007/s00401-010-0719-5). Moreover, the authors' observation, including perivascular lymphocytic infiltrate and fibrinoid degeneration of vessel walls with multinucleated giant cells, seems to differ from the parenchymal lymphocytic infiltrate around "collapsed plaques" and in the lack of frank CAARI and/or ABRA that were reported in cases of meningoencephalitis after AN1792 (DOI: 10.1111/j.1750-3639.2004.tb00493.x ; DOI: 10.1038/nm840). These arguments go against encephalitis and are instead in favor of CNS vasculitis. I understand that the nicely found acronym iCARE mainly depends on an E, hence encephalitis, but semantics matters, as underlined by the authors' proposition of acronym change. I would therefore advocate for the use of vasculitis or angiitis instead. Besides, this discussion about encephalitis vs. vasculitis could also be part of the article's discussion if the authors/editor deem it relevant.

2. Similar amyloid-related histiocytic vasculitis aspects were also recently reported under lecanemab in a patient who also received tPA (DOI: 10.1056/NEJMc2215148 ; DOI: 10.3233/JAD-221305). tPA is not known to induce such post-mortem vasculitic changes. Therefore, this case is worth mentioning here because it is neuropathologically very similar to this report. In addition, these 2 observations (the Chicago and the current case) could also allow the authors to discuss the highly improbable random nature of these vasculitic events under lecanemab that was argued by Eisai's consultants after the first case report (DOI: 10.1056/NEJMc2215907). We now have a minimum of 2 well-described observations of ABRA-like phenomena under lecanemab (2/1795 patients randomized in the CLARITY-AD phase 3 RCT). Spontaneous ABRA or CAA-related inflammation is infrequent (e.g., estimated to occur in about 0.13/100,000 individuals in Japan: DOI: 10.1111/ene.14031). Therefore, observation of ABRA-like phenomena at a minimum rate of ~1 per 900 treated patients is clearly abnormal; making the hypothesis of a random observation of spontaneous ABRA very unlikely. In my opinion, this is a strong argument in favor of a lecanemab-induced phenomenon that should be discussed in the current Manuscript.

3. In my opinion, the causality between lecanemab and the vasculitic event observed here could also be strengthened using IHC with anti-lecanemab antibodies. This use could allow for observing lecanemab around the vasculitic lesions and senile plaques. Contrasting the putative observation of lecanemab around vasculitic lesions vs. normal vessels could increase the level of evidence for causality between lecanemab and this ABRA-like phenomenon. It would also feed the discussion about the direct (lymphocyte/histiocytic activation) or indirect (excess perivascular clearance and BBB dysfunction) mechanistic effect of lecanemab on the vasculitic event observed here, and discussed by the authors. Similar approaches were already used to understand further the mechanism of gantenerumab in amyloid plaques clearance (DOI: 10.3233/JAD-2011-110977). I am unaware of commercially available anti-lecanemab antibodies, and it is unlikely that, if Eisai has some, they would agree to share them with the authors. I am also aware that the publication timing of this paper is important, and the lack of this information should not prevent its publication. However, it could be discussed as a limitation and perspective.

4. Previous autopsy reports from the AN1792 and aducanumab RCTs also highlighted the target engagement and plaque clearance induced by anti-amyloid mAbs, as well as their putative effect on tau pathology (DOI: 10.1007/s00401-022-02433-4 ; DOI: 10.1007/s00401-010-0719-5). The recently

published detailed neuropathological examination of the Chicago death under lecanemab also discusses the same aspects (DOI: 10.3233/JAD-221305). This should be mentioned and put into perspective.

5. As highlighted in the FDA briefing document regarding the Advisory Committee meeting for full approval of lecanemab (<https://www.fda.gov/media/169263/download> pp 26-27), there are a few discrepancies between the report by the author and Eisai's observations regarding the same patient. The major concern is the baseline number of microbleeds, which seemed to be null according to the centralized radiological reading performed by Eisai as planned in the RCT and four, according to the authors. In Figure 1B, the authors illustrate their own reading. Since microbleed reading can sometimes be challenging in the lack of upper or lower slides, could the authors better illustrate these 4 microbleeds in Figure 1B, using arrows?

6. The discussion about the relationship between CAA and ARIA is of utmost importance considering its implication for treated patients. This is, in my opinion, one of the major contributions of this case report by CAA experts. As illustrated by the FDA briefing document regarding the Advisory Committee meeting for full approval of lecanemab (<https://www.fda.gov/media/169263/download> p 34), this topic and the interpretation of current evidence remains highly debated and controversial: "any role for an interaction between lecanemab and underlying severe CAA or CAA-related inflammation/vasculitis cannot be determined. The two fatalities occurred in the OLE with no comparator control group. There is a high background prevalence of CAA in subjects with Alzheimer's disease, and a lack of definitive criteria for diagnosing CAA. This results in inability to compare the risk of cerebral hemorrhage in lecanemab-treated subjects with or without CAA and leads to substantial uncertainty in the ability to make any recommendations regarding use of lecanemab in subjects with CAA." Considering the authors' expertise in the CAA field, their opinion on this topic will be particularly considered. In its current form, I think this discussion could be improved.

a. "The risk of developing iCARE correlates with the presence of CAA and associated risk factors, including APOE4 genotype (10, 13). It seems reasonable to avoid treating patients carrying two copies of the E4 genotype with this medication." Unless I am mistaken, the references cited in this sentence do not refer to the link between ARIA and CAA. Since this is an important element of the argumentation, it should be corrected. To my best knowledge, the relationship between CAA and ARIA has yet to be well studied. Autopsy reports like this one, provide a mechanistic link between the two entities and reinforce the mechanistic hypotheses proposed a few years ago (DOI: 10.1038/s41582-019-0281-2). One of the authors was also a co-author of a recent commentary that reviewed the current evidence in that direction (imaging, genetics: DOI: 10.1161/STROKEAHA.121.03687), and recent reviews could also be mentioned (DOI: 10.1001/jamaneurol.2021.5205). This part of the article's discussion could be fleshed out more fully.

b. The discussion regarding current clinical CAA criteria is critical since, as underlined by the authors, the pre-lecanemab MRI of the patient fulfilled the Boston v2.0 criteria for probable CAA (> 50 yo + cognitive impairment + at least two strictly lobar cerebral microbleeds + white matter hyperintensities in a multispot pattern) but not the classical vascular MRI exclusion criteria used in these RCTs established by the AA working group (DOI: 10.1016/j.jalz.2011.05.2351). In this regard, the discussion sentence is unclear "In this case, the patient met diagnostic criteria for CAA. ": does it refer to the pre-lecanemab MRI or to the neuropath analysis? It should be detailed. Besides, the 2011 AA working group

recommendations to exclude individuals with more than 4 microbleeds without considering the topography was made considering the relationships between CAA and ARIA risk and the lobar topography of microbleeds in CAA: “It is recognized that substantial numbers of lobar [microhaemorrhages] mHs likely reflect the presence and severity of CAA, raising diagnostic and therapeutic considerations. Current prevalence estimates in mild to moderate AD are that 80% of patients with mH will have two or fewer mHs. Given the uncertainty of risk and concerns about CAA severity, the Workgroup supports the recommendation that the cutoff value of four mHs is used for exclusion in trials of amyloid-modifying therapies for AD. This threshold would allow the potential for imaging measurement variability to be taken into account and reflect the uncertainty regarding the clinical relevance of small numbers of mHs.” Therefore, the authors should further discuss (probably in line with more recent data as well as this case report) why the topography of microbleeds should now be considered.

c. The discussion around the choice of the MR sequence for microbleeds detection is also relevant regarding the observation they make and also regarding the sensitivity of these MR criteria in the evaluation of pre-lecanemab exclusion criteria, as also briefly mentioned in the Boston v2.0 criteria. Still, they claim the 2011 AA working group recommendations “do not standardize the screening MRI protocols”. This assertion is untrue since these guidelines clearly detailed the sequences to be used (section 19.1.1.1 from DOI: 10.1016/j.jalz.2011.05.2351). Moreover, it seems that there is some confusion in the description of the MR sequences used in the case report: the Fig.1 caption of in-vivo MRI mentions SWI (MIP) while they are described as T2 GRE in the Supplementary Material and that they do look like T2 GRE images to me. There are also other confusions in this regard in the text. This should be corrected throughout the Manuscript.

d. The question of CAA severity is unsolved but is likely important in this discussion of the relationship between ARIA and CAA. Indeed, CAA is observed in ~80% of AD-autopsied cases (DOI: 10.1002/alz.12366). Even if exclusion criteria used in these trials decreased this proportion, most individuals included in CLARITY-AD had likely neuropathologically-confirmed CAA; still, among those, only ~1% developed serious ARIA. So, it is very likely that most CLARITY AD participants with pathological evidence of CAA developed neither ARIA nor severe ARIA. This is important when proposing to rule out individuals with CAA, as the authors do in their conclusion sentence. Therefore, the authors should detail which Boston criteria they recommend (v1.5 or v2.0, which differ in terms of risk of recurrent intracerebral hemorrhage. doi: 10.1161/STROKEAHA.122.042407 and may also differ in regard to ARIA risk), and which sensitivity (possible vs. probable CAA?).

7. There is no mention of any patient’s legal representative consent. This should be disclosed.

8. From the Press, it sounded like the patient was admitted to a Floridian hospital (<https://www.science.org/content/article/scientists-tie-third-clinical-trial-death-experimental-alzheimer-s-drug>). I am surprised that only Floridian pathologists are amongst the authors, I assume intensivists or neurologists were also involved. Clinical and in-vivo MRI data detailed in this Case Report and collected within a Floridian hospital should be acknowledged.

9. Data presented in this Case Report seem to have been collected within the Eisai-sponsored CLARITY-

AD clinical trial (pre-lecanemab MRI, amyloid, and tau PET). I am not an expert in intellectual property for scientific publication, but the Editor and the Journal should verify that these data, though very valuable, can be published as is.

Minor concerns

1. In the Introduction, the authors mention “(CAARI) (which has sometimes been termed amyloid- β related angiitis or ABRA when the pathological features are particularly aggressive)”. My understanding of the neuropathological distinction between CAARI and ABRA relates to the observation of perivascular lymphocytic infiltrates (CAARI) vs. the observation of fibrinoid necrosis and multinucleated giant cells (ABRA). If correct, the Introduction sentence is relatively imprecise and should be reworded accordingly.
2. The following sentence logically connects CAARI and ARIA: “With passive antibody administration, a similar phenomenon has been observed and termed ARIA or β -amyloid related imaging abnormality.”. This remains debated, and this case report is actually one argument favoring a relationship between the 2 entities. Still, the wording should be more cautious and balanced in the Introduction.
3. Section “Brief case history”: “Neuroimaging revealed cerebral edema...”. The imaging technique and timing (detailed later in the manuscript) should be a little bit more detailed here: e.g. “Day 1 CT revealed..., which was confirmed on Day 3 MRI...”.
4. Was any CSF collection performed in this case? If yes, what were the results? If not, it should be briefly mentioned.
5. The precise date of the CTs and MRIs should be detailed in the Manuscript (as well as the date of the acute event).
6. Details regarding the MR scanners and sequences used to perform in vivo images should be detailed in the Manuscript.
7. Interpretation of the diffusion sequence should also indicate whether it was in favor of vasogenic or cytotoxic edema. It is indirectly mentioned and should be clearly stated.
8. The tau PET tracer used in the CLARITY AD RCT was flortaucipir (TAUVID™). I suppose this is also the case, here. This should be mentioned.
9. The name of the amyloid PET tracer is florbetaben, not fluorbetaben.
10. Are the pre-lecanemab infusion MMSE and CDR-SB of the patient available? If yes, it should be mentioned in the Manuscript.
11. Some of the 16 regions analyzed at autopsy seemed to be the regions evidencing edema without bleeding that were evidenced with the MRI. If yes, could the authors detail their observations regarding inflammation in these regions: parenchymal lymphocytic infiltrates? Vessel wall ruptures?
12. Was there any vasculitic observation in leptomeninges?
13. Was the meningioma observed at autopsy?
14. The sentence “These features will have considerable bearing on whether minor alterations in the cognitive trajectory of treated patients are likely to represent disease modification or purely symptomatic effects.” could be accompanied by this recent reference (DOI: 10.1001/jamaneurol.2023.0815).
15. The temporal poles symmetric WMHs seen on the pre-lecanemab MRI seem quite atypical to me, especially in a CAA and AD context. To my best knowledge, not a lot of neurological disorders give a similar pattern: CADASIL, Steinert, Fabry. Of course, there is an autopsy ensuring there is CAA and AD, and the clinical history does not favor these other diseases. Still, I am curious whether no other

comorbid vascular disease could explain this peculiar pattern. Is the PAS staining in the temporal poles normal? The pons WMHs are also atypical for CAA and AD. Is there evidence of arteriolosclerosis in the region at autopsy?

16. The exact age of this patient is 79, as mentioned in the abstract. It should be the same in the whole Manuscript (“aged between 75 and 80” should be corrected).

17. What medications and dosage were the patient taking for depression and gastroesophageal reflux when the fatal event occurred?

18. The date of the last lecanemab infusion and of the acute neurological event would be useful.

19. What was the initial and the follow-up Glasgow Coma Scale score?

20. Was the Day 2 tachycardia associated with troponin and liver enzyme elevation further investigated? Was it related to an ongoing acute coronary event or the AF? What was the physician’s hypothesis? Was any cardiac abnormality observed at the autopsy?

21. The authors determined the APOE status outside the RCT. Though there are several remaining mentions throughout the Manuscript that “it has not been yet released”. It should be removed from the Manuscript.

22. When was the EEG describing FIRDA performed?

23. Was the patient still partly sedated from Day 3 to death?

24. Typo in Suppl Mat, “Autopsy”: last sentence of the paragraph  “consistent with a homozygous E4 genotype”.

25. Supplementary Figures should have a more detailed caption detailing the abnormalities observed.

Reviewer #2 (Remarks to the Author):

The authors provide a very important clinical/radiological/pathological case study of an individual treated with Lecanemab who developed, and ultimately died from, a severe form of ARIA in the first month of receiving active treatment.

It is clearly important that cases such as these are in the public domain to allow for the mechanisms of ARIA to be evaluated, and for factors that might predispose to this rare, severe side-effect to be evaluated with the aim of determining exclusion criteria as/when the drug is made available outside of a clinical trial setting.

The report contains much of interest; the following suggestions are provided in an attempt to improve it further.

General points

I would suggest that the authors ensure that the report is factual and balanced and so would suggest that potentially pejorative words like “lethal”, “expired” and “culprit” are avoided.

Introduction

It should be stated that the 2002 study was AN1792, and that it was an active immunotherapy

“efflux capability OF the ...”

“The side-effect is common” ... this is true but it is important to put it into context. “Less severe” is too strong -- most cases are asymptomatic and this should be clearly stated here as it is later.

Brief case report

The abstract says she was 79, so I’m not sure why it’s 75-80 in the case history?

I think more detail is required in this “brief” version. What was the MMSE? What medications was she taking? The tau and amyloid scans were done prior to the open label study I think – what investigations were done prior to enrolment to the study? She presumably had an MRI. Were there microbleeds from the start?

It is important to state that the pre-active treatment scans showed significant white matter change and microbleeds – which in retrospect were compatible with CAA.

It is very important to state in this abbreviated version that in hospital she was in AF and was treated with heparin. It is not certain that this will have impacted on the outcome, but it is relevant.

Neuroimaging

The pre-treatment scan looks to me to show rather quite extensive white matter disease including within the brainstem. I think this should be commented on. I would have concerns about enrolling someone with this level of white matter disease in an anti-amyloid trial.

The microbleeds are not clearly see on Figure 1B and I think these should be highlighted with arrows on 1B and 1D.

It is Florbetaben not Fluorbetaben

What was the tau tracer?

Supplementary Figure 6 is important. The legend should describe what is seen in A (evidence for widespread cortical tracer retention in keeping with cortical amyloid deposition). I found Figure 6B uninterpretable and would suggest it be replaced and explained -- was the tau just in the MTL or more widespread?

It is not surprising that the PM imaging at higher field strength and following processing shows more bleeding than the in vivo scan. I would add “as expected”

Discussion

The authors variously used the terms vasculitis, encephalitis, meningo-encephalitis, and offer a new acronym – iCARE – when ARIA become symptomatic. I am not sure that adding a new term is helpful as it implies that all symptomatic ARIA is encephalitis – which may or not be the case. To my mind from a clinical perspective this is still ARIA, and it can be stated if this is asymptomatic, symptomatic, or in this case severe/fatal. What this report does very well is to highlight the pathological underpinnings, at least in this severe case.

Incidentally the lack of contrast enhancement on the scan is of interest and could be commented on?

While the authors state that it is reasonable to avoid treating patients who are E4 homozygous, it may be appropriate still if the dose is attenuated and they are closely monitored. This could perhaps be mentioned?

I found it notable that the patient had debilitating headaches after all three doses of Lecanemab before clearly deteriorating after dose 3. Could it be that severe headache should mandate scanning/dose halting until scanning and clinical assessment has been conducted? This is a potentially important clinical point that may prevent morbidity/mortality in the years to come.

They say that the patient fulfilled criteria for CAA. Presumably this was pre-treatment? In which case should CAA (e.g. Boston criteria) be a contra-indication rather than just number of micro bleeds?

The authors make the valid point that clinical MRI at lower field strengths may not pick up microbleeds. Ultimately, we will need clear criteria for who can/cannot be given these medications, and so it would be helpful if they - as experts in the field - could suggest protocols they recommend for in vivo use. SWI at 3T?

I agree with their recommendations about educating care providers about ARIA and the need for post-mortem studies in treated patients. In the spirit of balance, I think it would be helpful if they commented on the finding that there did seem to be clearance of amyloid from 21% of the plaques, and attenuation in a further 24%, and that these were the regions with most intense immunoreactivity for IBA1/microglia. While this was clearly a tragic case, it also provides evidence for the mechanism by which lecanemab clears amyloid.

Supplementary

The expanded case history in places rather repeats the abbreviated form. I would suggest including aspects of this in the abbreviated form (as above especially the AF/heparin) and omitting aspects which directly repeat what has already been stated. I would avoid some of the more colloquial language.

Reviewer #3 (Remarks to the Author):

This case history reflects the dangers associated with anti-amyloid monoclonal antibody treatments when Intramural Periarterial Drainage is compromised. My only comments are:

1. to introduce the exact age of the patient rather than a patient between age of 75 and 80;
2. to use the term Intramural Periarterial Drainage (IPAD) rather than perivascular clearance. As such, could the statement "The underlying pathophysiology of this side effect was felt to be due to rapid clearance of β -amyloid through the perivascular pathway overwhelming..." be changed to "The underlying pathophysiology of this side effect suggests that Intramural Periarterial Drainage along the walls of capillaries and arteries was overwhelmed.."
3. Consider removing the statement introducing a new term , as this is a case report, maybe introducing the term iatrogenic cerebral amyloid related encephalitis (iCARE) could be done in the context of a review on ARIA.

Response to reviews – response comments are in blue.

Reviewer 1:

The article “Fatal Iatrogenic Cerebral Amyloid-Related Encephalitis in a patient treated with lecanemab for Alzheimer’s disease” is a case report of a fatal complication of the recently FDA-approved anti-amyloid immunotherapy, lecanemab, as part of the Open-Label Extension of the pivotal phase 3 Eisai-sponsored randomized-controlled clinical trial (RCT). This case report from a very productive and well-renowned group in the field of cerebral amyloid angiopathy (CAA), includes a thorough neuropath examination, and clinical and neuroimaging data. The results are simply and clearly presented. The evidence is convincing, and the findings and the discussion are of utmost importance for the use of this drug in light of the current FDA discussion regarding full approval of lecanemab and its use in clinical practice, especially in APOE4 homozygous carriers and in individuals with CAA. Regarding the scientific and medical aspects, I would simply argue against the emphasis on the term encephalitis by the authors and propose the use of vasculitis or angiitis instead. I am not aware of the space and reference limitations, but I would also recommend a few additional details and discussion necessary to put these findings into perspective. I am particularly surprised that the authors do not mention the recent similar report of the Chicago group concerning a case of lecanemab-induced vasculitis (DOI: 10.1056/NEJMc2215148 ; DOI: 10.3233/JAD-221305). I am not an expert in legal and political aspects, but I also have some concerns. I am not aware of exactly which physicians were involved in the management of the patient. However, from the Press, it seemed that the acute clinical event detailed in this case report was managed in a Floridian hospital. I am surprised that only Floridian pathologists and not ICU physicians or neurologists involved in the management of the patient are co-authors of this paper. I understand that some political dimensions are probably at-stake. Nonetheless, the Editor and the Authors should clarify this, for transparency. Finally, some data seem to have been collected during the Eisai-sponsored RCT (pre-infusion MRI, Amyloid, and Tau PET). We are also not sure who the deceased patient's legal representative was, and who consented to this publication. The agreement of the patient legal representative and the intellectual property of these data should be carefully reviewed by the Journal’s legal department to ensure the article will not be retracted after publication despite its medical and scientific importance and rigor. Note that I have no neuropathological training and could not review these aspects of the current manuscript in detail.

We greatly appreciate the favorable reading from the reviewer and the extensive and detailed commentary which certainly improved the manuscript. This evaluation clearly took considerable time and effort, and we are grateful for the input. Detailed responses to each of the individual concerns are listed below.

Major concerns

1. I agree with the authors when they propose a new acronym and medical term to describe these complications under anti-amyloid immunotherapies: the previously proposed ARIA (Amyloid Related Imaging Abnormalities) tends to undermine the severity of these events. Nonetheless, the use of encephalitis is debatable and can be confusing since the neuropathological description of the fatal event is mainly that of a drug-induced central nervous system vasculitis/angiitis. Indeed, the parenchymal microglial activation observed here around plaques was previously reported in autopsies of patients treated with anti-amyloid drugs, but who did not experience ARIA (DOI: 10.1007/s00401-022-02433-4 ; DOI: 10.1007/s00401-010-0719-5). Moreover, the authors’ observation, including perivascular lymphocytic infiltrate and fibrinoid degeneration of vessel walls with multinucleated giant cells, seems to differ from the parenchymal lymphocytic infiltrate around “collapsed plaques” and in the lack of frank CAARI and/or ABRA that were reported in cases of meningoencephalitis after AN1792 (DOI: 10.1111/j.1750-3639.2004.tb00493.x ; DOI: 10.1038/nm840). These arguments go against encephalitis and are instead in favor of CNS vasculitis. I understand that the nicely found acronym iCARE mainly depends on an E, hence encephalitis, but semantics matters, as underlined by the authors’ proposition of acronym change. I would therefore advocate for the use of vasculitis or angiitis instead. Besides, this discussion about encephalitis vs. vasculitis could also be part of the article’s discussion if the authors/editor deem it relevant.

We thought carefully about this issue, and we feel strongly both about introducing a new term and that encephalitis is the correct designation. We hope we can persuade the reviewer with the following arguments and additional information.

The clinical and pathological findings in the inflammatory cases from the active immunization trial have been termed meningoencephalitis uncontested for over 20 years. The findings in our case and the recent case from Chicago are very similar to these earlier cases and occurred after treatment with a closely related treatment modality, so we would argue for the appropriateness of preserving the original terminology, that this is an encephalitis. The primary involvement of the vessels does not preclude the use of the term encephalitis; many encephalitides have a vascular predominance – for instance, HSV encephalitis has intense perivascular inflammatory infiltrates. Similarly in our case, the perivascular inflammation is associated with parenchymal inflammatory infiltrates as well. The inflammation in this case was not restricted to the vessels, although the vessels and perivascular spaces were the most notable areas of inflammation and the main focus of our discussion. We are adding additional images to Supplementary Figure 9 (and include below for reference) to illustrate that the CD68+ inflammatory cells extend into the parenchyma (which is not surprising).

Regarding the specific references the reviewer describes as lacking the vascular features shown in our case or in typical cases of CAARI -- in fact, these cases are quite comparable to both CAARI-spectrum pathology and to our current case.

The case in the 2004 *Brain Pathology* paper was shown to have perivascular lymphocytic infiltrates (figure 1A), cerebral amyloid angiopathy (figure 1g) and microvascular degeneration/microhemorrhage (figure 1l – the vessel wall is distended and separated into layers).

[REDACTED]

[Images at right taken from 2004 Brain Pathol paper, figure 1, rearranged to focus on key elements for this discussion.]

The case in the 2003 *Nature Medicine* paper, the MRI features right parieto-occipital edema/white matter changes (Fig 1a) and intense perivascular infiltrate of predominantly lymphocytes (Fig 3). Elements of fibrinoid degeneration were not commented on, but are clearly visible in the Fig3a. The presence of multinucleated giant cells are inconsistent in the CAARI/ABRA spectrum and could easily be affected by steroid treatments (a phenomenon which is well documented in other granulomatous vasculitides, see doi: 10.7326/0003-4819-120-12-199406150-00003), so their presence or absence should not greatly alter the interpretation of the pathology.

[REDACTED]

Images above taken from Figure 1 (left image) and Figure 3 (middle and right) of the 2003 Nature Med paper. The leftmost image shows focal cerebral edema in the right temporoparietal and occipital lobes. The right sided images show inflammatory leptomenigeal infiltrates are CD3+ (right). Images rearranged to focus on key elements for discussion.

The description of “collapsed plaques” in all four cited articles (the two previously mentioned, as well as James Nicoll’s 2010 series and Anita Huttner’s recent case in *Acta Neuropathol*) is consistent with examples we saw in this case, and we agree that this could be consistent with active β -amyloid clearance which may appear in cases without ARIA/iCARE.

We have expanded our discussion of these issues to try to more-thoroughly defend the use of the iCARE.

2. Similar amyloid-related histiocytic vasculitis aspects were also recently reported under lecanemab in a patient who also received tPA (DOI: 10.1056/NEJMc2215148 ; DOI: 10.3233/JAD-221305). tPA is not known to induce such post-mortem vasculitic changes. Therefore, this case is worth mentioning here because it is neuropathologically very similar to this report. In addition, these 2 observations (the Chicago and the current case) could also allow the authors to discuss the highly improbable random nature of these vasculitic events under lecanemab that was argued by Eisai’s consultants after the first case report (DOI: 10.1056/NEJMc2215907). We now have a minimum of 2 well-described observations of ABRA-like phenomena under lecanemab (2/1795 patients randomized in the CLARITY-AD phase 3 RCT). Spontaneous ABRA or CAA-related inflammation is infrequent (e.g., estimated to occur in about 0.13/100,000 individuals in Japan: DOI: 10.1111/ene.14031). Therefore, observation of ABRA-like phenomena at a minimum rate of ~ 1 per 900 treated patients is clearly abnormal; making the hypothesis of a random observation of spontaneous ABRA very unlikely. In my opinion, this is a strong argument in favor of a lecanemab-induced phenomenon that should be discussed in the current Manuscript.

We have added a discussion of the very relevant case report originating from Chicago and we have included a discussion regarding the rate of iCARE versus CAARI/ABRA as recommended by the reviewer which adds important perspective.

3. In my opinion, the causality between lecanemab and the vasculitic event observed here could also be strengthened using IHC with anti-lecanemab antibodies. This use could allow for observing lecanemab around the vasculitic lesions and senile plaques. Contrasting the putative observation of lecanemab around vasculitic lesions vs. normal vessels could increase the level of evidence for causality between lecanemab and this ABRA-like phenomenon. It would also feed the discussion about the direct (lymphocyte/histiocytic activation) or indirect (excess perivascular clearance and BBB dysfunction) mechanistic effect of lecanemab on the vasculitic event observed here, and discussed by the authors. Similar approaches were already used to understand further the mechanism of gantenerumab in amyloid plaques clearance (DOI: 10.3233/JAD-2011-110977). I am unaware of commercially available anti-lecanemab antibodies, and it is unlikely that, if Eisai has some, they would agree to share them with the authors. I am also aware that the publication timing of this paper is important, and the lack of this information should not prevent its publication. However, it could be discussed as a limitation and perspective.

We thought carefully about a range of approaches to try to demonstrate the presence of lecanemab within the abnormal tissue. This seemed to us to be technically difficult to accomplish with enough specificity to reach a definitive conclusion. Two specific points: 1) We were not convinced that showing lecanemab was present in the tissue materially altered the conversation regarding causality because the patient was in the acute phase of treatment with an immunotherapy which convincingly engages a brain target. We reasoned that the discovery of lecanemab in the tissue where the drug’s target resides would be expected whether or not there was ARIA/iCARE. The absence of lecanemab in the tissue might be of interest, but could be attributable to either the intensity of the inflammatory reaction or the

sensitivity of the assay, so again, we felt this result would not be robustly interpretable. 2) We are not certain that the referenced JAD article utilizes a methodology that would translate well in this setting. The authors of that study appear to have used an anti-human IgG1 secondary antibody to detect gantenerumab rather than an antibody specific for gantenerumab. In the context of the experiment, gantenerumab was placed on tissue sections as one would do with immunohistochemistry, then detected with an anti-human IgG1 secondary antibody – in the mice gantenerumab was likely the only human antibody for the secondary antibody to detect. In the human tissue, the specificity of the result likely came from the direct application of gantenerumab to the brain tissue which presumably is at a higher abundance than native human antibodies. The same situation would probably not be present in a patient treated intravenously. Moreover, specificity could not be assured because endogenous antibodies cross the blood brain barrier and enter the brain, and this would be exaggerated with the extent of edema and inflammation observed in this case. Consequently, we felt the results of such an assay would be difficult to interpret and add little to the manuscript.

4. Previous autopsy reports from the AN1792 and aducanumab RCTs also highlighted the target engagement and plaque clearance induced by anti-amyloid mAbs, as well as their putative effect on tau pathology (DOI: 10.1007/s00401-022-02433-4 ; DOI: 10.1007/s00401-010-0719-5). The recently published detailed neuropathological examination of the Chicago death under lecanemab also discusses the same aspects (DOI: 10.3233/JAD-221305). This should be mentioned and put into perspective.

We have added a discussion of these issues to the second to last paragraph of the discussion.

5. As highlighted in the FDA briefing document regarding the Advisory Committee meeting for full approval of lecanemab (<https://www.fda.gov/media/169263/download> pp 26-27), there are a few discrepancies between the report by the author and Eisai's observations regarding the same patient. The major concern is the baseline number of microbleeds, which seemed to be null according to the centralized radiological reading performed by Eisai as planned in the RCT and four, according to the authors. In Figure 1B, the authors illustrate their own reading. Since microbleed reading can sometimes be challenging in the lack of upper or lower slides, could the authors better illustrate these 4 microbleeds in Figure 1B, using arrows?

We have added the arrows as requested. We carefully re-reviewed the images after the discrepancy between our interpretation and Eisai's emerged. We feel a careful review of the imaging supports our interpretation.

6. The discussion about the relationship between CAA and ARIA is of utmost importance considering its implication for treated patients. This is, in my opinion, one of the major contributions of this case report by CAA experts. As illustrated by the FDA briefing document regarding the Advisory Committee meeting for full approval of lecanemab (<https://www.fda.gov/media/169263/download> p 34), this topic and the interpretation of current evidence remains highly debated and controversial: "any role for an interaction between lecanemab and underlying severe CAA or CAA-related inflammation/vasculitis cannot be determined. The two fatalities occurred in the OLE with no comparator control group. There is a high background prevalence of CAA in subjects with Alzheimer's disease, and a lack of definitive criteria for diagnosing CAA. This results in inability to compare the risk of cerebral hemorrhage in lecanemab-treated subjects with or without CAA and leads to substantial uncertainty in the ability to make any recommendations regarding use of lecanemab in subjects with CAA." Considering the authors' expertise in the CAA field, their opinion on this topic will be particularly considered. In its current form, I think this discussion could be improved.

a. "The risk of developing iCARE correlates with the presence of CAA and associated risk factors, including APOE4 genotype (10, 13). It seems reasonable to avoid treating patients carrying two copies of the E4 genotype with this medication." Unless I am mistaken, the references cited in this sentence do not refer to the link between ARIA and CAA. Since this is an important element of the argumentation, it should be corrected. To my best knowledge, the relationship between CAA and ARIA has yet to be well studied. Autopsy reports like this one, provide a mechanistic link between the two entities and reinforce the mechanistic hypotheses proposed a few years ago (DOI: 10.1038/s41582-019-0281-2). One of the authors was also a co-author of a recent commentary that reviewed the current evidence in that direction (imaging, genetics: DOI: 10.1161/STROKEAHA.121.03687), and recent reviews could also be mentioned (DOI: 10.1001/jamaneurol.2021.5205). This part of the article's discussion could be fleshed out more fully.

We appreciate the focus on improving the discussion. The Liu 2018 Neurology paper was cited to support the claim that ARIA-risk was linked to APOE4 status. The Salloway 2022 JAMA Neurology article was cited because it shows an increase in microhemorrhages with aducanumab treatment, that this association was stronger in APOE4 carriers and the association was strongest in those with ARIA-E. The link between CAA and ARIA/iCARE has been made by numerous previous writers (notably Hampel et al doi.org/10.1093/brain/awad188) and is not solely based on autopsy studies. MRI-based criteria are well-validated for CAA (microhemorrhages in a lobar pattern). Many anti-amyloid immunotherapy trials have witnessed worsening of lobar microhemorrhage burden. Moreover, vascular reactivity has been assessed pre- and post- anti-amyloid immunotherapy and was shown to worsen with treatment, further supporting a physiological link with CAA (doi.org/10.1002/acn3.761). And, while we agree that the neuropathological link between ARIA/iCARE and CAA needs to be strengthened, the available evidence supports the assertion that it is linked to severe CAA. We have added a discussion of these details to this section.

b. The discussion regarding current clinical CAA criteria is critical since, as underlined by the authors, the pre-lecanemab MRI of the patient fulfilled the Boston v2.0 criteria for probable CAA (> 50 yo + cognitive impairment + at least two strictly lobar cerebral microbleeds + white matter hyperintensities in a multispot pattern) but not the classical vascular MRI exclusion criteria used in these RCTs established by the AA working group (DOI: 10.1016/j.jalz.2011.05.2351). In this regard, the discussion sentence is unclear “In this case, the patient met diagnostic criteria for CAA. “: does it refer to the pre-lecanemab MRI or to the neuropath analysis? It should be detailed. Besides, the 2011 AA working group recommendations to exclude individuals with more than 4 microbleeds without considering the topography was made considering the relationships between CAA and ARIA risk and the lobar topography of microbleeds in CAA: “It is recognized that substantial numbers of lobar [microhaemorrhages] mHs likely reflect the presence and severity of CAA, raising diagnostic and therapeutic considerations. Current prevalence estimates in mild to moderate AD are that 80% of patients with mH will have two or fewer mHs. Given the uncertainty of risk and concerns about CAA severity, the Workgroup supports the recommendation that the cutoff value of four mHs is used for exclusion in trials of amyloid-modifying therapies for AD. This threshold would allow the potential for imaging measurement variability to be taken into account and reflect the uncertainty regarding the clinical relevance of small numbers of mHs.” Therefore, the authors should further discuss (probably in line with more recent data as well as this case report) why the topography of microbleeds should now be considered.

The patient’s pre-treatment MRI meets criteria for probable CAA based on any version of the Boston criteria. In contrast, the criteria used in the clinical trial are not based on validated diagnostic data. We emphasize more clearly why the topography of microhemorrhages is important and also advocate to use validated diagnostic criteria rather than the current expert opinion-based recommendations from the Alzheimer’s Association working group.

c. The discussion around the choice of the MR sequence for microbleeds detection is also relevant regarding the observation they make and also regarding the sensitivity of these MR criteria in the evaluation of pre-lecanemab exclusion criteria, as also briefly mentioned in the Boston v2.0 criteria. Still, they claim the 2011 AA working group recommendations “do not standardize the screening MRI protocols”. This assertion is untrue since these guidelines clearly detailed the sequences to be used (section 19.1.1.1 from DOI: 10.1016/j.jalz.2011.05.2351). Moreover, it seems that there is some confusion in the description of the MR sequences used in the case report: the Fig.1 caption of in-vivo MRI mentions SWI (MIP) while they are described as T2 GRE in the Supplementary Material and that they do look like T2 GRE images to me. There are also other confusions in this regard in the text. This should be corrected throughout the Manuscript.

The reference to the criteria not standardizing MRI protocols is a reference to the clinical trial exclusion criteria not the recommendations from the Working group (the sentence before this one is “The trial exclusion criteria permit enrollment of patients with up to 4 microhemorrhages”) – the field strength and other critical imaging parameters were not specified in the study protocol. We did not include much discussion of the imaging criteria proposed in the 2011 Workgroup document because they essentially advocated for lowest-common denominator settings and omitted discussion of some key variables (they proposed 1.5 Telsa minimum field strength and 5mm maximum slice thickness with any iron sensitive sequence, they did not address interslice gap). The statement said that the primary aim of using

these criteria was that the studies could be performed on a broad range of scanners in any care environment, whereas we would prioritize using the most appropriate settings for detecting clinically relevant microhemorrhages. As a consequence of these non-restrictive standards, the range of sensitivities of MRIs associated with these clinical trials is variable. This patient was assessed with GRE-T2* at 3T with a 0.5mm interslice gap, meaning almost 10% of the brain was not imaged. We have added these imaging parameters to the appropriate figure legend.

d. The question of CAA severity is unsolved but is likely important in this discussion of the relationship between ARIA and CAA. Indeed, CAA is observed in ~80% of AD-autopsied cases (DOI: 10.1002/alz.12366). Even if exclusion criteria used in these trials decreased this proportion, most individuals included in CLARITY-AD had likely neuropathologically-confirmed CAA; still, among those, only ~1% developed serious ARIA. So, it is very likely that most CLARITY AD participants with pathological evidence of CAA developed neither ARIA nor severe ARIA. This is important when proposing to rule out individuals with CAA, as the authors do in their conclusion sentence. Therefore, the authors should detail which Boston criteria they recommend (v1.5 or v2.0, which differ in terms of risk of recurrent intracerebral hemorrhage. doi: 10.1161/STROKEAHA.122.042407 and may also differ in regard to ARIA risk), and which sensitivity (possible vs. probable CAA?).

The discrepancy between neuropathological series and radiological series reporting the prevalence of CAA is primarily because most cases in neuropathological series of CAA have mild CAA, but microhemorrhages correspond to moderate and severe CAA on neuropathological evaluation (doi.org/10.1212/WNL.46.6.1592). Based on neuropathology to neuropsychiatric phenotyping correlations, mild CAA does not have a clear neuropsychological correlate and it is likely that mild CAA represents physiological clearance of β -amyloid from the brain in most cases rather than a pathological process (doi.org/10.1212/WNL.0000000000002175). The Boston criteria is based on the presence of MRI detectable microvascular degeneration and is unlikely to detect most cases of mild CAA, but performs well diagnostically (especially with SWI) for clinically relevant levels of CAA, which are present in roughly 20-30% of patients with AD (DOI: 10.1016/S1474-4422(22)00208-3). MRI-detectable CAA is associated with APOE4 and roughly parallels the frequency of ARIA in the clinical trials. Importantly, every single published case of ARIA/iCARE that has been evaluated neuropathologically not only has CAA, but has severe CAA. We recommend using the newest Boston Criteria (2.0) for diagnosing probable CAA with a 3T scan using susceptibility weighted imaging and no interslice gap and have added this to the discussion.

7. There is no mention of any patient's legal representative consent. This should be disclosed.

This was included in the Acknowledgments section. We have added a reference to the informed consent to the main text as well.

8. From the Press, it sounded like the patient was admitted to a Floridian hospital (<https://www.science.org/content/article/scientists-tie-third-clinical-trial-death-experimental-alzheimer-s-drug>). I am surprised that only Floridian pathologists are amongst the authors, I assume intensivists or neurologists were also involved. Clinical and in-vivo MRI data detailed in this Case Report and collected within a Floridian hospital should be acknowledged.

9. Data presented in this Case Report seem to have been collected within the Eisai-sponsored CLARITY-AD clinical trial (pre-lecanemab MRI, amyloid, and tau PET). I am not an expert in intellectual property for scientific publication, but the Editor and the Journal should verify that these data, though very valuable, can be published as is.

Re: 8 & 9. Everyone who met criteria for authorship is included as an author. We have provided a detailed response to the editors on this issue. Briefly, there is an extensive precedent for publishing this type of material and we feel the importance to ongoing discussions around patient safety take priority. The material was legally obtained from the patient's family and we see no barrier to proceeding. If the editors have ongoing concerns, the manuscript remains effective without reproducing images from the pre-lecanemab MRI and the PET scans, and they could be omitted.

Minor concerns

1. In the Introduction, the authors mention “(CAARI) (which has sometimes been termed amyloid- β related angiitis or ABRA when the pathological features are particularly aggressive)”. My understanding of the neuropathological distinction between CAARI and ABRA relates to the observation of perivascular lymphocytic infiltrates (CAARI) vs. the observation of fibrinoid necrosis and multinucleated giant cells (ABRA). If correct, the Introduction sentence is relatively imprecise and should be reworded accordingly.

The diagnostic differences between CAARI and ABRA are imprecise and often due to sampling of a patchy pathology and/or the effect of treatments (often corticosteroids). In my view, CAARI and ABRA should be treated as a single entity because they behave as a single entity from a clinical and neuroimaging standpoint. Both have extensive microhemorrhagic changes on imaging, so clearly involvement of the vessel wall is present in both, whether it is visualized pathologically or not. (We are not alone in this view – this article from a panel of experts similar treats the conditions as more or less synonymous: doi.org/10.1093/brain/awad188). Nevertheless, we do not want to engage with this issue in the manuscript at greater length because it is tangential to the main point, so we have removed the parenthetical statement.

2. The following sentence logically connects CAARI and ARIA: “With passive antibody administration, a similar phenomenon has been observed and termed ARIA or β -amyloid related imaging abnormality.”. This remains debated, and this case report is actually one argument favoring a relationship between the 2 entities. Still, the wording should be more cautious and balanced in the Introduction.

We clarify here that the clinical and neuroimaging features of ARIA are similar to that of CAARI – which is well established.

3. Section “Brief case history”: “Neuroimaging revealed cerebral edema...”. The imaging technique and timing (detailed later in the manuscript) should be a little bit more detailed here: e.g. “Day 1 CT revealed..., which was confirmed on Day 3 MRI...”.

We added these details, as requested.

4. Was any CSF collection performed in this case? If yes, what were the results? If not, it should be briefly mentioned.

CSF was not analyzed.

5. The precise date of the CTs and MRIs should be detailed in the Manuscript (as well as the date of the acute event).

The publication of precise dates may constitute identifiable patient information is not permitted by our IRB protocol.

6. Details regarding the MR scanners and sequences used to perform in vivo images should be detailed in the Manuscript.

These have been added.

7. Interpretation of the diffusion sequence should also indicate whether it was in favor of vasogenic or cytotoxic edema. It is indirectly mentioned and should be clearly stated.

We have added that the edema was vasogenic.

8. The tau PET tracer used in the CLARITY AD RCT was flortaucipir (TAUVID™). I suppose this is also the case, here. This should be mentioned.

We have added this detail.

9. The name of the amyloid PET tracer is florbetaben, not fluorbetaben.

Corrected.

10. Are the pre-lecanemab infusion MMSE and CDR-SB of the patient available? If yes, it should be mentioned in the Manuscript.

These are not available, but detailed descriptions of her functional status were provided by family and study partner – these details are included.

11. Some of the 16 regions analyzed at autopsy seemed to be the regions evidencing edema without bleeding that were evidenced with the MRI. If yes, could the authors detail their observations regarding inflammation in these regions: parenchymal lymphocytic infiltrates? Vessel wall ruptures?

We have images from the frontal lobe in areas that are mildly affected and areas which seem unaffected. In unaffected areas, there are many plaques and CAA, but minimal perivascular hypercellularity. As you would expect, in areas with

mild edema without hemorrhage, we did not see hemorrhages microscopically. The inflammatory changes were present as in the more-severely affected areas, but less severe. We will add a comment about this to the manuscript and have included a few images below to illustrate.

12. Was there any vasculitic observation in leptomeninges?

Yes, the added images in supplementary figure 9 demonstrate inflammatory changes in the leptomeninges. We have also added an additional video showing a wider field of view of the vascular degeneration from a cleared specimen. This shows how extensive the microvascular degeneration is, including leptomeningeal, grey matter and white matter vessels.

13. Was the meningioma observed at autopsy?

The meningioma was not evaluated during the autopsy. It was very small and stable over many years of clinical follow-up (it is visible in the GRE images in figure 1B&D).

14. The sentence “These features will have considerable bearing on whether minor alterations in the cognitive trajectory of treated patients are likely to represent disease modification or purely symptomatic effects.” could be accompanied by this recent reference (DOI: 10.1001/jamaneurol.2023.0815).

We agree and have added the reference.

15. The temporal poles symmetric WMHs seen on the pre-lecanemab MRI seem quite atypical to me, especially in a CAA and AD context. To my best knowledge, not a lot of neurological disorders give a similar pattern: CADASIL, Steinert,

Fabry. Of course, there is an autopsy ensuring there is CAA and AD, and the clinical history does not favor these other diseases. Still, I am curious whether no other comorbid vascular disease could explain this peculiar pattern. Is the PAS staining in the temporal poles normal? The pons WMHs are also atypical for CAA and AD. Is there evidence of arteriosclerosis in the region at autopsy?

I agree the temporal pole white matter disease is remarkable is interesting, but at age 79 with no family history and minimal symptoms, these genetic disorders are extremely unlikely and we did not pursue them. In addition to cerebral amyloid angiopathy, she may have had undiagnosed hypertension to account for the findings. We have added a comment that her general autopsy found evidence of cardiac hypertrophy which was interpreted as most consistent with chronic hypertension.

16. The exact age of this patient is 79, as mentioned in the abstract. It should be the same in the whole Manuscript (“aged between 75 and 80” should be corrected).

This has been corrected.

17. What medications and dosage were the patient taking for depression and gastroesophageal reflux when the fatal event occurred?

We have added the medications to the supplementary history. She was on few medications, none of which seemed relevant.

18. The date of the last lecanemab infusion and of the acute neurological event would be useful.

The publication of precise dates may constitute identifiable patient information is not permitted by our IRB protocol.

19. What was the initial and the follow-up Glasgow Coma Scale score?

In the acute phase, the EMS team documented that her GCS score was <8. Her clinical status after the sedative and paralytic agent wore off improved: she was mute and hemiparetic, but awake and interacting with her environment (GCS was not formally documented). We now report this information as it was recorded in her medical record.

20. Was the Day 2 tachycardia associated with troponin and liver enzyme elevation further investigated? Was it related to an ongoing acute coronary event or the AF? What was the physician’s hypothesis? Was any cardiac abnormality observed at the autopsy?

It was interpreted as occurring secondary to rapid atrial fibrillation with a possible contribution from the cerebral edema. She was found to have moderate coronary artery disease and left ventricular hypertrophy on autopsy, which was interpreted as most consistent with chronic hypertension (although she did not have a history of clinically overt hypertension).

21. The authors determined the APOE status outside the RCT. Though there are several remaining mentions throughout the Manuscript that “it has not been yet released”. It should be removed from the Manuscript.

They have been removed.

22. When was the EEG describing FIRDA performed?

It was done on admission day 1. We have added the timing of the EEG to the report.

23. Was the patient still partly sedated from Day 3 to death?

During that period of time, she received small doses of lorazepam for anxiolysis to assist with diagnostic testing, 25mg BID quetiapine and intermittent dexmedetomidine infusions to address agitation. Once her goals of care shifted to comfort oriented measures, additional sedatives were used.

24. Typo in Suppl Mat, “Autopsy”: last sentence of the paragraph  “consistent with a homozygous E4 genotype”.

Corrected, thanks.

25. Supplementary Figures should have a more detailed caption detailing the abnormalities observed.

We have worked on this.

Reviewer #2:

The authors provide a very important clinical/radiological/pathological case study of an individual treated with Lecanemab who developed, and ultimately died from, a severe form of ARIA in the first month of receiving active treatment. It is clearly important that cases such as these are in the public domain to allow for the mechanisms of ARIA to be evaluated, and for factors that might predispose to this rare, severe side-effect to be evaluated with the aim of determining exclusion criteria as/when the drug is made available outside of a clinical trial setting. The report contains much of interest; the following suggestions are provided in an attempt to improve it further.

Many thanks for this evaluation and the detailed points that followed. As mentioned to the first reviewer, this clearly represents a significant time investment, and we are grateful for the care effort that went into the comments.

General points

I would suggest that the authors ensure that the report is factual and balanced and so would suggest that potentially pejorative words like “lethal”, “expired” and “culprit” are avoided.

We don't necessarily agree that any of these terms are inherently pejorative, nor that they were used in a pejorative fashion, but we have substituted alternatives for each.

Introduction

1. It should be stated that the 2002 study was AN1792, and that it was an active immunotherapy
2. “efflux capability OF the ...”
3. “The side-effect is common” ... this is true but it is important to put it into context. “Less severe” is too strong -- most cases are asymptomatic and this should be clearly stated here as it is later.

All adjusted as recommended by the reviewer.

Brief case report

The abstract says she was 79, so I'm not sure why it's 75-80 in the case history?

We will report her actual age, this has been updated.

I think more detail is required in this “brief” version. What was the MMSE? What medications was she taking? The tau and amyloid scans were done prior to the open label study I think – what investigations were done prior to enrolment to the study? She presumably had an MRI. Were there microbleeds from the start? It is important to state that the pre-active treatment scans showed significant white matter change and microbleeds – which in retrospect were compatible with CAA.

We do not have access to her imaging pre-enrollment in the placebo-controlled study, only prior to the open label extension. As she was randomized to the placebo arm, the pre-OLE images can be considered baseline studies because the study protocol required re-evaluation of entry and exclusion criteria at the onset of the OLE. Results of cognitive testing are not available for publication, but detailed descriptions of her functional status were provided by family and her study partner and these details are included. We have added her medications to the extended clinical history in the supplement, none were felt to be major players in her acute event. We included a statement that she met criteria for CAA in the original draft – we have tried to amplify this statement in the revision to make it clearer.

It is very important to state in this abbreviated version that in hospital she was in AF and was treated with heparin. It is not certain that this will have impacted on the outcome, but it is relevant.

We have added this to the brief history. Importantly, the heparin was initiated a significant period after her acute decline, then was paused again once the diagnosis of ARIA was considered. She had no macrohemorrhage.

Neuroimaging

The pre-treatment scan looks to me to show rather quite extensive white matter disease including within the brainstem. I think this should be commented on. I would have concerns about enrolling someone with this level of white matter disease in an anti-amyloid trial.

We agree. The exclusion criteria was for “severe white matter disease”, and while the criteria for severe white matter disease was not defined, we suspect that by some clinical criteria or expert opinion this could be considered severe. The exclusion criteria in most of the anti-amyloid immunotherapy trials (including this one) don’t specify a specific scale for assessing WMH, but the aducanumab trials used a score of 3 on the ARWMC scale

(<https://www.ahajournals.org/doi/10.1161/01.str.32.6.1318>). By this measure, the patient would probably have scored a 2 (she has more than 1 basal ganglia lesion, but WMH are not confluent in the basal ganglia / she has the “beginnings of confluent lesions” in subcortical zones, but there is not “diffuse involvement of the entire region”) thus would have qualified for enrollment. We have added a comment to this point.

The microbleeds are not clearly see on Figure 1B and I think these should be highlighted with arrows on 1B and 1D.

It is Florbetaben not Fluorbetaben

What was the tau tracer?

All added or corrected as requested.

Supplementary Figure 6 is important. The legend should describe what is seen in A (evidence for widespread cortical tracer retention in keeping with cortical amyloid deposition). I found Figure 6B uninterpretable and would suggest it be replaced and explained -- was the tau just in the MTL or more widespread?

We have expanded this legend.

It is not surprising that the PM imaging at higher field strength and following processing shows more bleeding than the in vivo scan. I would add “as expected”

We agree. Added.

Discussion

The authors variously used the terms vasculitis, encephalitis, meningo-encephalitis, and offer a new acronym – iCARE – when ARIA become symptomatic. I am not sure that adding a new term is helpful as it implies that all symptomatic ARIA is encephalitis – which may or not be the case. To my mind from a clinical perspective this is still ARIA, and it can be stated if this is asymptomatic, symptomatic, or in this case severe/fatal. What this report does very well is to highlight the pathological underpinnings, at least in this severe case.

ARIA is fundamentally an incorrect term because it implies that this process is an imaging abnormality, not a clinically important syndrome. Once symptomatic or once the mass effect and/or bleeding is severe even if asymptomatic, we feel the term ARIA is no longer applicable. The history of term is unambiguous that it was meant to downplay the severity of this side effect from anti-amyloid immunotherapies, and that needs to be corrected. We will defer to the editors, but we advocate strongly to introduce this, more correct, term. A more extended defense of the appropriateness of the term iCARE is included in the response to reviewer 1.

Incidentally the lack of contrast enhancement on the scan is of interest and could be commented on?

We agree that this is interesting - we have added a comment to the neuroimaging section highlighting this. Patients with CAARI occasionally have contrast enhancement, but most do not. The details of the approach to imaging may be

relevant here (longer delays between contrast administration and image acquisition will permit a longer time for contrast extravasation and therefore greater sensitivity).

While the authors state that it is reasonable to avoid treating patients who are E4 homozygous, it may be appropriate still if the dose is attenuated and they are closely monitored. This could perhaps be mentioned?

We understand that there are a range of thoughtful views on this topic, but we authors don't agree with the reviewer here. The benefits to patients with APOE4/4 appear to have been less than to E3 counterparts and the risks are higher, so in our view it would be reasonable to avoid treating this subgroup. The recent publication of the donanemab data further supports this view as the patients with APOE4/4 again benefited least. We have clarified that our position is not only based on the safety concern, but also a lack of efficacy in this subgroup.

I found it notable that the patient had debilitating headaches after all three doses of Lecanemab before clearly deteriorating after dose 3. Could it be that severe headache should mandate scanning/dose halting until scanning and clinical assessment has been conducted? This is a potentially important clinical point that may prevent morbidity/mortality in the years to come.

It's an important question – certainly in this case the family was impressed by the headaches which were quite out of the ordinary for her and timed closely with her infusions. It is unclear from the available clinical trial data how often the headaches reported as adverse events were similarly closely timed to infusions and whether this was associated with a higher risk of ARIA. We raise this question in the revised manuscript (2nd paragraph of the discussion).

They say that the patient fulfilled criteria for CAA. Presumably this was pre-treatment? In which case should CAA (e.g. Boston criteria) be a contra-indication rather than just number of micro bleeds?

Yes, the reference is to the pre-treatment MRI. And yes, we would advocate for any patient meeting Boston criteria (2.0) for probable CAA to be excluded from treatment.

The authors make the valid point that clinical MRI at lower field strengths may not pick up microbleeds. Ultimately, we will need clear criteria for who can/cannot be given these medications, and so it would be helpful if they - as experts in the field - could suggest protocols they recommend for in vivo use. SWI at 3T?

Yes, to enhance the detection of microhemorrhages, we would advocate to use SWI at 3T with no interslice gap, slice thickness of 5mm or less as the standard criteria for the initial scan.

I agree with their recommendations about educating care providers about ARIA and the need for post-mortem studies in treated patients. In the spirit of balance, I think it would be helpful if they commented on the finding that there did seem to be clearance of amyloid from 21% of the plaques, and attenuation in a further 24%, and that these were the regions with most intense immunoreactivity for IBA1/microglia. While this was clearly a tragic case, it also provides evidence for the mechanism by which lecanemab clears amyloid.

Agree. We have added additional emphasis on this issue in the first paragraph of the discussion.

Supplementary

The expanded case history in places rather repeats the abbreviated form. I would suggest including aspects of this in the abbreviated form (as above especially the AF/heparin) and omitting aspects which directly repeat what has already been stated. I would avoid some of the more colloquial language.

We have edited this passage (also in accordance with the comments from reviewer 1).

Reviewer #3:

This case history reflects the dangers associated with anti-amyloid monoclonal antibody treatments when Intramural

Periarterial Drainage is compromised. My only comments are:

1. to introduce the exact age of the patient rather than a patient between age of 75 and 80;

Agree, we have corrected this.

2. to use the term Intramural Periarterial Drainage (IPAD) rather than perivascular clearance. As such, could the statement "The underlying pathophysiology of this side effect was felt to be due to rapid clearance of β -amyloid through the perivascular pathway overwhelming..." be changed to "The underlying pathophysiology of this side effect suggests that Intramural Periarterial Drainage along the walls of capillaries and arteries was overwhelmed."

Agree, we have made this adjustment.

3. Consider removing the statement introducing a new term, as this is a case report, maybe introducing the term iatrogenic cerebral amyloid related encephalitis (iCARE) could be done in the context of a review on ARIA.

We appreciate this point of view and again will submit to the preferences of the editors, but for the reasons indicated to the other two reviewers, we feel strongly about introducing this more-accurate term.

REVIEWERS' COMMENTS

Reviewer #1 (Remarks to the Author):

The authors have made significant efforts to address the points raised by the reviewers. They have provided detailed and satisfactory responses to most of the concerns. Therefore, I believe that the manuscript is suitable for publication in Nature Communication except for one issue. This issue, which was also raised by the other two reviewers, concerns using the term "meningo-encephalitis." I will explain my reasoning against it in the detailed review below. Additionally, though this point didn't strike me during my first review, a comment from Reviewer 2 made me realize that the Tau PET scan in Supplementary Figure 6b shows neocortical tau aggregates, while the neuropathological examination classified it as Braak stage IV. These two pieces of information seem incompatible since neocortical Tau PET signal is always associated with Braak VI stage at autopsy. I suggest that this point needs clarification before publication.

Major concerns:

1. Use of the term meningoencephalitis and acronym iCARE (=reply to the previous 1st major concern of my 1st review, to which the authors answered). Black is my 1st review, blue the authors' answer and red my current response (see enclosed pdf file for colors):

"1. I agree with the authors when they propose a new acronym and medical term to describe these complications under anti-amyloid immunotherapies: the previously proposed ARIA (Amyloid Related Imaging Abnormalities) tends to undermine the severity of these events. Nonetheless, the use of encephalitis is debatable and can be confusing since the neuropathological description of the fatal event is mainly that of a drug-induced central nervous system vasculitis/angiitis. Indeed, the parenchymal microglial activation observed here around plaques was previously reported in autopsies of patients treated with anti-amyloid drugs, but who did not experience ARIA (DOI: 10.1007/s00401-022- 02433-4 ; DOI: 10.1007/s00401-010-0719-5). Moreover, the authors' observation, including perivascular lymphocytic infiltrate and fibrinoid degeneration of vessel walls with multinucleated giant cells, seems to differ from the parenchymal lymphocytic infiltrate around "collapsed plaques" and in the lack of frank CAARI and/or ABRA that were reported in cases of meningoencephalitis after AN1792 (DOI: 10.1111/j.1750-3639.2004.tb00493.x ; DOI: 10.1038/nm840). These arguments go against encephalitis and are instead in favor of CNS vasculitis. I understand that the nicely found acronym iCARE mainly depends on an E, hence encephalitis, but semantics matters, as underlined by the authors' proposition of acronym change. I would therefore advocate for the use of vasculitis or angiitis instead. Besides, this discussion about encephalitis vs. vasculitis could also be part of the article's discussion if the authors/editor deem it relevant.

We thought carefully about this issue, and we feel strongly both about introducing a new term and that encephalitis is the correct designation. We hope we can persuade the reviewer with the following arguments and additional information. The clinical and pathological findings in the inflammatory cases from the active immunization trial have been termed meningoencephalitis uncontested for over 20 years. The findings in our case and the recent case from Chicago are very similar to these earlier cases

and occurred after treatment with a closely related treatment modality, so we would argue for the appropriateness of preserving the original terminology, that this is an encephalitis. The primary involvement of the vessels does not preclude the use of the term encephalitis; many encephalitides have a vascular predominance – for instance, HSV encephalitis has intense perivascular inflammatory infiltrates. Similarly in our case, the perivascular inflammation is associated with parenchymal inflammatory infiltrates as well. The inflammation in this case was not restricted to the vessels, although the vessels and perivascular spaces were the most notable areas of inflammation and the main focus of our discussion. We are adding additional images to Supplementary Figure 9 (and include below for reference) to illustrate that the CD68+ inflammatory cells extend into the parenchyma (which is not surprising). Regarding the specific references the reviewer describes as lacking the vascular features shown in our case or in typical cases of CAARI -- in fact, these cases are quite comparable to both CAARI-spectrum pathology and to our current case. The case in the 2004 Brain Pathology paper was shown to have perivascular lymphocytic infiltrates (figure 1A), cerebral amyloid angiopathy (figure 1g) and microvascular degeneration/microhemorrhage (figure 1l – the vessel wall is distended and separated into layers). [Images at right taken from 2004 Brain Pathol paper, figure 1, rearranged to focus on key elements for this discussion.] The case in the 2003 Nature Medicine paper, the MRI features right parieto-occipital edema/white matter changes (Fig 1a) and intense perivascular infiltrate of predominantly lymphocytes (Fig 3). Elements of fibrinoid degeneration were not commented on, but are clearly visible in the Fig3a. The presence of multinucleated giant cells are inconsistent in the CAARI/ABRA spectrum and could easily be affected by steroid treatments (a phenomenon which is well documented in other granulomatous vasculitides, see doi: 10.7326/0003-4819-120-12-199406150-00003), so their presence or absence should not greatly alter the interpretation of the pathology. The description of “collapsed plaques” in all four cited articles (the two previously mentioned, as well as James Nicoll’s 2010 series and Anita Huttner’s recent case in Acta Neuropathol) is consistent with examples we saw in this case, and we agree that this could be consistent with active β -amyloid clearance which may appear in cases without ARIA/iCARE. We have expanded our discussion of these issues to try to more-thoroughly defend the use of the iCARE.”

This reviewer took the initiative to seek the help of a neuropathologist who specializes in neuroinflammation and neuroinfectious diseases, to provide a detailed response. This reviewer wrote a reply, which was then corrected by the neuropathologist specialized in neuroinflammation. The authors of the research paper agree that the findings fall within the spectrum of CAARI/ABRA, as they state in the Discussion section of the manuscript: "Neuropathologically, we found that her condition was associated with marked perivascular inflammation and arteriolar degeneration resembling fibrinoid necrosis, leading to microhemorrhagic changes both in the parenchyma and leptomeninges. The inflammatory features include extensive macrophages and/or activated microglia in the leptomeninges, perivascular space and adjacent parenchyma, with occasional multinucleated giant cells in vessel walls. These features were associated with severe cerebral amyloid angiopathy. The patient’s neuroimaging, along with this constellation of neuropathology findings, is similar to the sporadic condition cerebral amyloid angiopathy related inflammation (CAARI).”

This phenomenon is widely accepted as an angiitis/vascularitis phenomenon, as exemplified by the ABRA acronym. Additionally, Supplementary Figures 9a and 9c confirm lymphocytic infiltrates within the vessel walls. The radiological aspect described here, without parenchymal contrast enhancement, also aligns

with this view. The additional images provided in Supplementary Figure 9 do not demonstrate a parenchymal lymphocytic infiltrate but a microglial activation that extends to the parenchyma. This is a frequently observed phenomenon in vasculitides.

It is important to note that the present study reports key inflammatory differences compared to the meningo-encephalitides found in AN1792 trials (<https://www.nature.com/articles/nm840>). The case reports from those trials showed massive and diffuse leptomeningeal and parenchymal infiltrates of lymphocytes (Fig 3 e-f-g-h from Nicoll et al., 2003 previously cited). These infiltrates were not limited to areas near blood vessels. However, in this study, the authors observed only perivascular lymphocytic phenomena and parenchymal microglial activation. Therefore, it cannot be concluded that the inflammation phenomenon observed is entirely similar to the meningo-encephalitides under the AN1792 trials. Hence, to ensure clarity, the term meningo-encephalitis should be replaced with angiitis or vasculitis.

While strictly speaking, any encephalon inflammation phenomenon could be qualified as encephalitis, including vasculitis, the term meningo-encephalitis could lead to confusion. The iCARE acronym could be replaced with iCAR - Iatrogenic Cerebral Amyloid-Related angiitis/vasculitis or iCARE but with artEritis instead of Encephalitis, or any other creative suggestion such as iaTrogenic cerEbral Amyloid Related vaSculitis – TEARS or iatroGenic cerebRal amyloid related vascuLitis - GRILL.

2. Supplementary Figure 6b illustrates the flortaucipir PET scan of this patient during the CLARITY-AD trial. The image indicates the presence of tau aggregates in the neocortex, specifically in the temporal and parietal cortices. However, the autopsy report mentions that the patient had a Braak IV stage, which suggests the absence of neurofibrillary tangles (NFT) in the parietal cortex. Studies comparing flortaucipir to neuropathology have consistently reported that the former can only detect tau aggregates in Braak stages \geq IV or V. This observation is supported by my personal clinical experience and the literature (doi:10.1001/jamaneurol.2020.0528 and DOI:10.1093/brain/awaa276). It seems unlikely that an unarguably neocortical positive flortaucipir PET scan corresponds to a post-mortem Braak IV stage. Therefore, the authors should verify both images and neuropathological Braak staging to clear up any confusion. If this is indeed a genuine finding, it should be discussed thoroughly in the main manuscript.

Minor concerns:

1. Neuroimaging section of the Main Manuscript, line 115. If correct (see major concern point 2), the tau positivity should be detailed as “neocortical tau positivity”
2. Neuropathology section of the Main Manuscript, line 132. ADNC should also be expressed as Intermediate according to the cited Hyman 2012 framework.
3. The Manuscript (main or supplementary) should mention that no CSF analysis was performed.
4. Details on the medications received during the hospital stay should be added to the Suppl Mat (reply to Minor concern nb 23)

Typos:

1. Introduction, line 62: AN1892  AN1792
2. Brief case history, line 84: Alzheimer’s Disease  Alzheimer’s disease

Reviewer #3 (Remarks to the Author):

The authors responded carefully and adequately to all comments.

The authors have made significant efforts to address the points raised by the reviewers. They have provided detailed and satisfactory responses to most of the concerns. Therefore, I believe that the manuscript is suitable for publication in *Nature Communication* except for one issue. This issue, which was also raised by the other two reviewers, concerns using the term "meningo-encephalitis." I will explain my reasoning against it in the detailed review below. Additionally, though this point didn't strike me during my first review, a comment from Reviewer 2 made me realize that the Tau PET scan in Supplementary Figure 6b shows neocortical tau aggregates, while the neuropathological examination classified it as Braak stage IV. These two pieces of information seem incompatible since neocortical Tau PET signal is always associated with Braak VI stage at autopsy. I suggest that this point needs clarification before publication.

We thank the reviewer for their efforts aimed at improving this case report and are happy to hear that the manuscript is overall fit to be published in *Nature Communications*. In our point-by-point response, we've addressed the major concerns 1 and 2 as well as the minor concerns.

Major concerns:

1. Use of the term meningoencephalitis and acronym iCARE (=reply to the previous 1st major concern of my 1st review, to which the authors answered). Black is my 1st review, blue the authors' answer and red my current response:

"1. I agree with the authors when they propose a new acronym and medical term to describe these complications under anti-amyloid immunotherapies: the previously proposed ARIA (Amyloid Related Imaging Abnormalities) tends to undermine the severity of these events. Nonetheless, the use of encephalitis is debatable and can be confusing since the neuropathological description of the fatal event is mainly that of a drug-induced central nervous system vasculitis/angiitis. Indeed, the parenchymal microglial activation observed here around plaques was previously reported in autopsies of patients treated with anti-amyloid drugs, but who did not experience ARIA (DOI: 10.1007/s00401-022-02433-4 ; DOI: 10.1007/s00401-010-0719-5). Moreover, the authors' observation, including perivascular lymphocytic infiltrate and fibrinoid degeneration of vessel walls with multinucleated giant cells, seems to differ from the parenchymal lymphocytic infiltrate around "collapsed plaques" and in the lack of frank CAARI and/or ABRA that were reported in cases of meningoencephalitis after AN1792 (DOI: 10.1111/j.1750-3639.2004.tb00493.x ; DOI: 10.1038/nm840). These arguments go against encephalitis and are instead in favor of CNS vasculitis. I understand that the nicely found acronym iCARE mainly depends on an E, hence encephalitis, but semantics matters, as underlined by the authors' proposition of acronym change. I would therefore advocate for the use of vasculitis or angiitis instead. Besides, this discussion about encephalitis vs. vasculitis could also be part of the article's discussion if the authors/editor deem it relevant.

We thought carefully about this issue, and we feel strongly both about introducing a new term and that encephalitis is the correct designation. We hope we can persuade the reviewer with the following arguments and additional information. The clinical and pathological findings in the inflammatory cases from the active immunization trial have been termed meningoencephalitis uncontested for over 20 years. The findings in our case and the recent case from Chicago are very similar to these earlier cases and occurred after treatment with a closely related treatment modality, so we would argue for the appropriateness of preserving the original terminology, that this is an encephalitis. The primary involvement of the vessels does not preclude the use of the term encephalitis; many encephalitides have a vascular predominance – for instance, HSV encephalitis has intense perivascular inflammatory infiltrates. Similarly in our case, the perivascular inflammation is associated with

parenchymal inflammatory infiltrates as well. The inflammation in this case was not restricted to the vessels, although the vessels and perivascular spaces were the most notable areas of inflammation and the main focus of our discussion. We are adding additional images to Supplementary Figure 9 (and include below for reference) to illustrate that the CD68+ inflammatory cells extend into the parenchyma (which is not surprising). Regarding the specific references the reviewer

describes as lacking the vascular features shown in our case or in typical cases of CAARI -- in fact, these cases are quite comparable to both CAARI- spectrum pathology and to our current case. The case in the 2004 Brain Pathology paper was shown to have perivascular lymphocytic infiltrates (figure 1A), cerebral amyloid angiopathy (figure 1g) and microvascular degeneration/microhemorrhage (figure 1l – the vessel wall is distended and separated into layers). [Images at right taken from 2004 Brain Pathol paper, figure 1, rearranged to focus on key elements for this discussion.] The case in the 2003 Nature Medicine paper, the MRI features right parieto-occipital edema/white matter changes (Fig 1a) and intense perivascular infiltrate of predominantly lymphocytes (Fig 3). Elements of fibrinoid degeneration were not commented on, but are clearly visible in the Fig3a. The presence of multinucleated giant cells are inconsistent in the CAARI/ABRA spectrum and could easily be affected by steroid treatments (a phenomenon which is well documented in other granulomatous vasculitides, see doi: 10.7326/0003-4819-120-12-199406150-00003), so their presence or absence should not greatly alter the interpretation of the pathology.

The description of “collapsed plaques” in all four cited articles (the two previously mentioned, as well as James Nicoll’s 2010 series and Anita Huttner’s recent case in Acta Neuropathol) is consistent with examples we saw in this case, and we agree that this could be consistent with active β -amyloid clearance which may appear in cases without ARIA/iCARE. We have expanded our discussion of these issues to try to more-thoroughly defend the use of the iCARE.”

This reviewer took the initiative to seek the help of a neuropathologist who specializes in neuroinflammation and neuroinfectious diseases, to provide a detailed response. This reviewer wrote a reply, which was then corrected by the neuropathologist specialized in neuroinflammation. The authors of the research paper agree that the findings fall within the spectrum of CAARI/ABRA, as they state in the Discussion section of the manuscript: "Neuropathologically, we found that her condition was associated with marked perivascular inflammation and arteriolar degeneration resembling fibrinoid necrosis, leading to microhemorrhagic changes both in the parenchyma and leptomeninges. The inflammatory features include extensive macrophages and/or activated microglia in the leptomeninges, perivascular space and adjacent parenchyma, with occasional multinucleated giant cells in vessel walls. These features were associated with severe cerebral amyloid angiopathy. The patient’s neuroimaging, along with this constellation of neuropathology findings, is similar to the sporadic condition cerebral amyloid angiopathy related inflammation (CAARI).”

This phenomenon is widely accepted as an angiitis/vascularitis phenomenon, as exemplified by the ABRA acronym. Additionally, Supplementary Figures 9a and 9c confirm lymphocytic infiltrates within the vessel walls. The radiological aspect described here, without parenchymal contrast enhancement, also aligns with this view. The additional images provided in Supplementary Figure 9 do not demonstrate a parenchymal lymphocytic infiltrate but a microglial activation that extends to the parenchyma. This is a frequently observed phenomenon in vasculitides.

It is important to note that the present study reports key inflammatory differences compared to the meningo-encephalitides found in AN1792 trials (<https://www.nature.com/articles/nm840>). The case reports from those trials showed massive and diffuse leptomeningeal and parenchymal infiltrates of lymphocytes (Fig 3 e-f-g-h from Nicoll et al., 2003 previously cited). These infiltrates were not limited to areas near blood vessels. However, in this study, the authors observed only perivascular lymphocytic phenomena and parenchymal microglial activation. Therefore, it cannot be concluded that the inflammation phenomenon observed is entirely similar to the meningo-encephalitides under the AN1792 trials. Hence, to ensure clarity, the term meningo-encephalitis should be replaced with angiitis or vasculitis.

While strictly speaking, any encephalon inflammation phenomenon could be qualified as encephalitis, including vasculitis, the term meningo-encephalitis could lead to confusion. The iCARE acronym could be

replaced with iCAR - Iatrogenic Cerebral Amyloid-Related angiitis/vasculitis or iCARE but with arteritis instead of Encephalitis, or any other creative suggestion such as iatrogenic cerebral Amyloid Related vasculitis – TEARS or iatrogenic cerebral amyloid related vasculitis - GRILL.

We now refer to this condition as an arteritis, as the reviewer requests, and have removed the iCARE acronym.

2. Supplementary Figure 6b illustrates the flortaucipir PET scan of this patient during the CLARITY-AD trial. The image indicates the presence of tau aggregates in the neocortex, specifically in the temporal and parietal cortices. However, the autopsy report mentions that the patient had a Braak IV stage, which suggests the absence of neurofibrillary tangles (NFT) in the parietal cortex. Studies comparing flortaucipir to neuropathology have consistently reported that the former can only detect tau aggregates in Braak stages \geq IV or V. This observation is supported by my personal clinical experience and the literature (doi:10.1001/jamaneurol.2020.0528 and DOI:10.1093/brain/awaa276). It seems unlikely that an unarguably neocortical positive flortaucipir PET scan corresponds to a post-mortem Braak IV stage. Therefore, the authors should verify both images and neuropathological Braak staging to clear up any confusion. If this is indeed a genuine finding, it should be discussed thoroughly in the main manuscript.

We thank the reviewer for the attentive review of our manuscript. We confirmed the accuracy of our neuropathological assessment.

The reviewer mentioned that PET detection of tau aggregates is limited to Braak stages \geq IV or V. Our case report concerns a woman who had Braak stage IV, which satisfies this criterion (Braak stages \geq IV or V), mentioned by reviewer. Additionally, we have carefully reviewed the two articles that the reviewer cited as evidence of incompatibility of Braak stage IV and positive flortaucipir PET scan.

[REDACTED]

1. Here is a relevant excerpt from the article in *JAMA Neurology* by Fleisher *et al.* (2020). As you can see, about 8% of the patients with Braak IV demonstrated Advanced AD which the authors explain as including neocortical tracer uptake in the parietal or frontal lobes (indicated by green box for your convenience). Furthermore, about 20% of the patients with Braak IV demonstrated moderate AD tau load (increase neocortical uptake in the posterolateral temporal or occipital regions).

2. Here is an excerpt from Figure 7 in the article in *Brain* by Soleimani-Meigooni *et. al* (2020).

[REDACTED]

This article gives a couple of examples of patients with Braak stages < IV showing elevated tau on the flortaucipir PET scan:

- “Patient 4, who had primary PSP pathology and contributing Alzheimer’s disease pathology [intermediate ADNC (A3, B2, C2), Braak III] showed elevated FTP-PET SUVR”.
- “ Patient 8, who had primary FTLT-tau pathology due to MAPT S305I showed elevated FTP-PET SUVR in the entorhinal cortex region of interest, but this patient did not have any NFT pathology”.

Furthermore, as you can see in the blue box in the Figure 7, there was only one patient with Braak stage IV in the entire study. Thus, no conclusions can be made about the ability of flortaucipir to detect NFT at Braak stage IV when only one person with Braak stage IV is considered.

We hope that this provides sufficient further clarification, which was requested by the reviewer. We have noted neocortical tau-tracer uptake in this revision of the text.

Minor concerns:

1. **Neuroimaging section of the Main Manuscript, line 115. If correct (see major concern point 2), the tau positivity should be detailed as “neocortical tau positivity”**

As requested, we have added this term and replied to the major concern 2 in detail.

2. **Neuropathology section of the Main Manuscript, line 132. ADNC should also be expressed as Intermediate according to the cited Hyman 2012 framework.**

Added "intermediate" to the ABC scores in the Neuropathology section of the main manuscript.

3. **The Manuscript (main or supplementary) should mention that no CSF analysis was performed.**

We added a statement regarding the fact that no CSF analysis was performed into the Supplement, line 47.

4. **Details on the medications received during the hospital stay should be added to the Suppl Mat (reply to Minor concern nb 23)**

We have added a thorough discussion of the medications the patient was administered to the supplementary clinical history. We have included all sedating drugs in the case history.

Typos:

5. Introduction, line 62: AN1892→AN1792

Corrected, thank you.

6. Brief case history, line 84: Alzheimer's Disease→Alzheimer's disease

Corrected as well.